# Research on quantitative evaluation of digital economy policy in China based on the PMC index model

Shuai Hong[1,2◉], Tianzun Wang[1,3◉]*, Xiaoyi Fu[1‡], Guo Li[1‡]

**1** Institute of Economic Research, Hebei University of Economics and Business, Shijiazhuang, China, **2** Hebei Coordinated Innovation Center for BTH Coordinated Development, Hebei University of Economics and Business, Shijiazhuang, China, **3** College of Economics and Management, Beijing University of Technology, Beijing, China

◉ These authors contributed equally to this work.
‡ XF and GL also contributed equally to this work.
* 15933601367@163.com

**Data Availability Statement:** All relevant data are within the paper and its Supporting Information files.

**Funding:** The National Social Science Foundation of China, 21BJL073, A/Prof. Xuebin Tian. The

## Abstract

The development of digital economy is a strategic choice to grasp the revolution of new science and technology and the new opportunities of industrial reform. The development of digital economy depends on the good support of policy and theoretical system. Therefore, the quantitative evaluation of policy texts provides the basis of decision-making and the suggestions of path optimization for the formulation and improvement of digital economy policy of China. By selecting the text of digital economy policy issued by China government, the paper constructs a quantitative evaluation model of digital economy policy using the methods of content analysis and text mining. The empirical research results show that the overall design evaluation of the selected policy is relatively reasonable. Six policies were evaluated as excellent and two as acceptable. In view of the problems such as lack of predictive policy in the policy type, lack of specific policy in the policy timeliness, imbalance in the use of policy guarantee, and lack of comprehensive coverage in the policy objectives, the paper puts forward corresponding countermeasures and suggestions.

## Introduction

As a new economic form, digital economy plays a significant role in driving industrial reform and enabling high-quality development. It is an important embodiment of national comprehensive strength in the digital age and an important engine for building a modern economic system. Influenced by the revolution of new science and technology and industrial reform, the development situation of digital economy of China is undergoing profound changes, and the trend of digital transformation is inevitable. Meanwhile, many countries have issued strategic plans one after another. The purpose is to create new competitive advantages and reshape the new international pattern in the digital era. From the perspective of theoretical logic and practical experience, building a new development pattern of digital economy requires the guidance

funder had no role in study design, data collection and analysis, decision to publish, or preparation of the manuscript.

**Competing interests:** The authors have declared that no competing interests exist.

and support of relevant policies under the background of industrial reform. At present, the research on digital economy policy of China mainly focuses on policy evolution, path optimization and enterprise innovation. However, there are few researches on digital economy from the perspective of policy text quantification.

Digital economy refers to high-tech development, business and social transformation, and information changes that drive economic growth. Inadequate funding of information and computing infrastructure may be a significant challenge in the transformation of the digital economy [1]. Improving the efficiency and volume of foreign investment is the effective way to promote the digital economy [2], and strengthen the application of key technologies of digital economy in specific regions [3]. With the acceleration of the digital development process, developing digital infrastructure and expanding the scope of digital applications will be conducive to the digital economy [4]. In fact, the new and old features of the digital economy are accurately grasped, which is conducive to opening up new market space [5]. Digital economy will greatly reduce market friction, but it also poses the new challenges to the effective operation of the market [6]. National policy priorities for digital economy development need to be determined and the countries need to increase support for the digital transformation process [7]. Furthermore, the countries need to consider all aspects of formulating cluster policies, which include the types of authoritarian and liberal [8]. There is clear evidence indicating the importance of digital flow, but enterprises have cleverly covered up the digital flow, while national policies and rules have failed to produce effective responses so far [9]. The competition policy in the digital economy should be based on sound theory and strict evidence analysis, and it is best to summarize it with the method of law and economics [10]. The digital economy is changing the competitive conditions of the digital market, and the new conditions of market competition and the new business models pose problems for regulators [11]. In addition, the growth of digital economic aggravates digital exclusion, inequality, unfavorable integration and other digital hazards [12]. Exploring policy options to alleviate a series of new challenges emerging in the digital economy and implementing precise policies, are the important tasks for coping with the Corona Virus Disease 2019 (COVID-19) and achieving sustainable economic development [13, 14].

The theory of public policy evaluation is a theoretical framework for assessing policies, aimed at determining their effectiveness, feasibility, sustainability, and fairness. Its scope of application encompasses various domains, including government policies, social policies, and economic policies. On one hand, this theory enhances the scientific and practical aspects of policy formulation by offering a scientific and systematic approach to evaluating policy effects. On the other hand, evaluation outcomes assist policymakers in comprehending the effects of policy implementation, thereby enabling policy adjustments and improvements based on the evaluation results. Existing methods for public policy evaluation each possess distinctive characteristics, providing crucial references for assessments related to digital economic policies. Due to the distinct scopes of application for each method, evaluation outcomes also exhibit slight variations. The grey relational degree model exhibits subjectivity and substantial error in determining optimal values for various indicators. The slow convergence of the Back Propagation (BP) neural network hampers the precision of policy evaluation. The fuzzy comprehensive evaluation method lacks objectivity in assessing indicators. It's noteworthy that current policy evaluation research excessively emphasizes ex-post evaluation, paying insufficient attention to the policies themselves. In contrast, the Policy Modeling Consistency (PMC) index model showcases a more prominent comprehensive evaluation effect. This method underscores the internal consistency of policies, offering advantages such as diversity assessment, pliability of indicators, and distinguishable gradations. Additionally, serving as a model for evaluating policy texts, it boasts traits like low cost and ease of operation, which mitigate subjectivity during

the evaluation process. While assessing the merits and demerits of individual policies, it better reveals policy disparities across different periods or regions through policy comparisons.

The PMC index model has a very wide range of applications, mainly embodied in regional sustainable development policy [15], green development policy [16], new energy vehicle industry policy [17], carbon emissions and carbon neutral policy [18], manufacturing development policy [19], watershed ecological compensation policy [20], public health emergency policy [21], waste classification management policy [22], disaster relief policy [23], pig price regulation policy [24], etc. It involves the fields of development, land, environmental protection, energy, industry, ecology, health, waste, disaster relief, and livelihood. Furthermore, relevant research focuses on policy timeliness, policy issuing institutions, policy audience scope, policy type, policy intensity, policy social benefits, policy incentives and constraints, etc. The overall design, improvement space, optimization path, theoretical expansion, decision-making basis, countermeasures and suggestions, management practice, technical support and evaluation of corresponding policies will be discussed. To sum up, it is very necessary and valuable to use the PMC index model to conduct policy evaluation on digital economy of China.

Compared with other methods, the PMC index model has important advantages. The evaluation dimensions are rich, and by adding evaluation dimensions instead of calculating indicator weights, it effectively avoids indicator weight errors and subjective evaluation biases, making the evaluation results more objective and accurate. Focusing on pre-policy evaluation, analyzing key content and keywords in policy texts, fills the gap in research and analysis of policy content. The PMC index evaluation method can be applied to both single industry policies and regional policy systems.

The research objective of this paper is to employ the PMC index model's research methodology to provide an in-depth interpretation of China's digital economic policies. Building upon text mining and content analysis, this study employs a combined quantitative and qualitative approach to thoroughly explore the strengths and weaknesses of China's digital economic policies. Subsequently, it presents corresponding optimization paths and recommendations, aiming to furnish scientific decision-making foundations for expediting the development of China's digital economic market.

The digital economy policy is a series of policy tool combinations that promote the development of the digital economy, involving multiple dimensions such as policy nature, policy objectives, policy content, policy effects, etc. How to objectively evaluate is a very important issue. At present, there is relatively little literature on the evaluation of digital economy policies, and policy evaluation is the most important link in the process of public policy formulation and management [25, 26]. It uses relevant research methods to systematically measure and judge the effectiveness of policy intervention and implementation [27]. Through the evaluation of digital economy policies, not only can scientific judgments be made on the value of the policies themselves [28], but also the actual effects of policy formulation and implementation can be tested [29]. Therefore, in view of this, this article combines digital economy policy sample analysis, text mining, and PMC index model to construct a quantitative evaluation index system for digital economy policies. It conducts sample analysis and text mining on China's digital economy policies, and quantitatively evaluates and analyzes typical digital economy policy texts at the central and local levels, in order to provide decision-making basis for the improvement of relevant policies and the formulation of new policies.

The contributions of this study are as follows:

Quantitative assessment of national and provincial-level digital economic policy texts in China is conducted through the utilization of content analysis and text mining methods. This endeavor aims to gain deeper insights into the ongoing trends within China's digital economic industry. Moreover, it seeks to furnish optimized recommendations for the formulation and

refinement of future policies. Simultaneously, from an empirical standpoint, a quantitative evaluation of China's digital economic policies is performed. The outcomes of policy evaluation serve as a reference for standardizing and enhancing the practicality of the future digital economic industry policy evaluation system, thereby facilitating the high-quality development of China's digital economic sector.

## Materials and methods

### Data source

Any single indicator can be misleading, but if multiple composite indicators point to the same result, it can provide a more accurate judgment of the evaluated thing. The PMC index first determines the meaning and level of variables at all levels, and then evaluates and analyzes the policy's advantages and disadvantages through the aggregated consistency level. This method attempts to find highly saturated secondary variables that can characterize policy characteristics and assign consistent weights to these variables to avoid subjective limitations. In addition, all variables are binary balanced, greatly simplifying the complexity of PMC index calculation.

This article uses a composite analysis method that combines policy sample analysis, text mining, and PMC index model. On the basis of sorting out China's digital economy policies, text mining is used to search for important and highly correlated text data, which forms a component of secondary indicators. Unstructured text data is transformed into structured and readable data, and the PMC index model is used to conduct quantitative evaluation research on digital economy policies in central and local areas of China.

The selection criteria for digital economic policies are as follows: To ensure the comprehensiveness and authority of the policy sample content, the focus is primarily on national and provincial-level digital economic policy texts. National-level policies, being overarching documents, provide stronger guidance and standardization, serving as crucial foundations for provincial policy formulation, guiding and constraining provincial policies. The selection of provincial policies considers their coordination with the central government and effectiveness. City-level and county-level policies are often extensions and supplements to provincial policies, and therefore, they are not included in the sample selection.

The initial policy search is conducted using the "PKU Law" (Peking University Legal Information Retrieval System) legal professional database with the title "Digital Economy," specifying the policy category as currently effective, and the retrieval date as January 1, 2023. Informal decision documents such as "letters" and "administrative licensing approvals" are excluded, focusing on formal decision documents like laws, regulations, resolutions, orders, opinions, and notifications. To ensure accuracy, verification and supplementation are performed on the official websites of the central government and various provincial governments. Finally, a manual full-text review is employed to eliminate policies with little relevance to the research theme. In total, 37 policy texts were retrieved (detailed information in S1 and S2 Appendices), including 5 national-level policies and 32 provincial-level policies. The main distribution results are presented in Table 1.

Table 1. Distribution of digital economy policies.

| Policy Level | National Policies | | Provincial Policies | |
|---|---|---|---|---|
| Policy Quantity | Administrative Regulations | Departmental Regulations | Local Regulations | Local Regulatory Documents |
| | 1 | 4 | 6 | 26 |

## Model construction

**Variable classification and parameter identification.** With the help of the text mining software ROSTCM.6, the 37 selected policy texts related to the digital economy were processed to extract representative words from them. Since the policy content pertains to the digital economy, it is necessary to manually remove high-frequency words such as "digital" and "economy," as well as adverbs and verbs that have no obvious effect on the results, such as "development," and so on. After eliminating the aforementioned words, 60 high-frequency effective words were finally identified, as shown in Table 2. This provides an important reference for setting secondary variables.

Adhering to the modeling principles of the PMC index model and grounded in the "Omnia Mobilis" [30] hypothesis, which posits interconnectivity among entities while not disregarding any existing variables, equal weights are assigned to both primary and secondary variables.

Compared with the traditional economy, the digital economy focuses on the application of products and the extension of services, is demand oriented, focuses on discovering potential and intangible user needs, provides personalized services, and creates user value. Due to the existence of digital technology, more consumers are involved in the production and consumption of products, so the dominant position of the digital economy is relatively unclear.

The digital economy is essentially a technology economy paradigm, which is an optimal practice model for economy and society based on technological innovation. It responds to structural crises caused by technological changes through institutional changes, thus forming a relatively stable and sustainable behavior. The digital economy is driven by digital knowledge and information as key production factors, modern information networks as the main carrier, and the efficient utilization of information and communication technology as an important driving force for efficiency improvement and economic optimization. Therefore, introducing data elements and changing social production methods can provide an important experimental environment for expanding current economic and management theories; The development of the digital economy requires new theories from economics and management to explain and promote the construction of new theories.

Since China first included the digital economy in the Two Sessions Report in 2017, the national level has attached increasing importance to promoting the development of the digital economy. To address various practical issues in the development of the digital economy, various Chinese ministries have issued a series of policies and regulations. The introduction of numerous policies has also made the policy and regulatory system in this field increasingly complex, leading to ineffective implementation and poor coordination in the policy integration environment. Policy analysis is the main basis for the abolition, reform, and establishment of digital economy policies. The primary task of policy analysis is to identify and conceptualize the problems that need to be solved, and policy problems stem from unresolved practical problems within the current policy system. The evaluation and analysis based on digital economy policy texts have important practical value and significance. Therefore, when quantitatively evaluating China's digital economy policies, this article will focus on distinguishing the advantages and disadvantages of current policies based on existing practical problems, providing decision-making basis for the improvement of relevant policies and the formulation of new policies.

Employing Ruiz's [31] research methodology, the 10 primary variables for the variable setting in the PMC model of digital economy policy have been established. These 10 primary variables are as follows: Policy Type ($X_1$), Policy Effectiveness ($X_2$), Policy Level ($X_3$), Policy Felids ($X_4$), Policy Guarantee ($X_5$), Policy Audience ($X_6$), Policy Objectives ($X_7$), Policy Evaluation ($X_8$), Policy Perspective ($X_9$), and Policy Publicity ($X_{10}$). Concurrently, through text mining

**Table 2. Statistics of effective vocabulary and word frequency.**

| NO. | Word | Frequency | NO. | Word | Frequency | NO. | Word | Frequency |
|---|---|---|---|---|---|---|---|---|
| 1 | Data | 2416 | 21 | Intelligence | 704 | 41 | Legal | 448 |
| 2 | Department | 2264 | 22 | Wisdom | 652 | 42 | Communication | 448 |
| 3 | Service | 2232 | 23 | Market | 624 | 43 | Fuse | 444 |
| 4 | Build | 1984 | 24 | Government | 616 | 44 | Overall | 436 |
| 5 | Digitization | 1640 | 25 | Society | 604 | 45 | Cooperation | 432 |
| 6 | Technology | 1584 | 26 | Guarantee | 604 | 46 | Ability | 416 |
| 7 | Enterprise | 1420 | 27 | Transformation | 580 | 47 | Director | 392 |
| 8 | Platform | 1372 | 28 | Information | 580 | 48 | Agriculture | 388 |
| 9 | Facilities | 1232 | 29 | Mechanism | 556 | 49 | Protect | 380 |
| 10 | Innovate | 1220 | 30 | Internet | 536 | 50 | Organization | 380 |
| 11 | Basics | 1212 | 31 | Coordination | 520 | 51 | Standard | 364 |
| 12 | Administration | 1196 | 32 | Science | 492 | 52 | Education | 352 |
| 13 | Application | 1140 | 33 | Level | 492 | 53 | Local | 348 |
| 14 | Public | 1076 | 34 | Supervise | 488 | 54 | Reform | 348 |
| 15 | Resources | 1072 | 35 | Cultivation | 484 | 55 | Law | 328 |
| 16 | Industry | 980 | 36 | Mechanism | 484 | 56 | Open | 328 |
| 17 | Field | 852 | 37 | Plan | 464 | 57 | Core | 324 |
| 18 | Share | 780 | 38 | Develop | 460 | 58 | Factor | 324 |
| 19 | System | 760 | 39 | Pattern | 460 | 59 | Employment | 320 |
| 20 | Security | 708 | 40 | Govern | 456 | 60 | Talent | 304 |

and considering the current state of digital economic development, and drawing insights from the research of scholars Kuang [32], Liu [33], Hu [34] and Yang [35], the Chinese digital economic policy PMC model variables were formulated. This encompasses 10 primary variables and 45 secondary variables, and the evaluation criteria are in binary form, with detailed outcomes presented in Table 3. The main procedural steps are as follows: assigning values to policy data based on corresponding indicators, inputting these values into a multi-input-output table, calculating the PMC index for policy texts, and subsequently plotting the PMC surface graph.

**Table of multiple-input-output.** The parameter configuration for the PMC index model primarily employs a binary approach, assuming that the importance of each secondary variable for input-output is equal, thereby effectively considering each variable. When the expressions regarding the policy under evaluation align with the corresponding evaluation criteria of a given secondary variable, that secondary variable is assigned a value of 1; conversely, it is assigned 0. This approach ensures that every variable is adequately accounted for. Combined with the specific conditions for the variables of the PMC index model of digital economic policy of China, the table of multi-input-output is established, as shown in Table 4.

## Empirical study on policy evaluation

### Selection of evaluation objects

The index model of PMC is designed to objectively consider all secondary variables without any special requirements for the evaluation object. It conducts quantitative evaluation on any digital economic policy, but subjective deviation should be minimized when selecting policy samples to be evaluated [36]. Therefore, in this study, a simple random sampling method was employed to select 8 policies for evaluation from the pool of 37 digital economic policies. They

**Table 3. Variable setting of PMC model of digital economy policy.**

| Primary Variables | Code | Secondary Variables | Code | Evaluation Criterion |
|---|---|---|---|---|
| Policy Type | $X_1$ | Forecast | $X_{1:1}$ | Whether it is predictive, Yes for 1, No for 0. |
| | | Supervise | $X_{1:2}$ | Whether supervision is involved, Yes for 1, No for 0. |
| | | Proposal | $X_{1:3}$ | Whether countermeasures and suggestions are included, Yes for 1, No for 0. |
| | | Describe | $X_{1:4}$ | Whether there is descriptive content, Yes for 1, No for 0. |
| | | Guide | $X_{1:5}$ | Whether it is instructive, Yes for 1, No for 0. |
| Policy Effectiveness | $X_2$ | Long-term | $X_{2:1}$ | Whether the content involves for more than 5 years, Yes for 1, No for 0. |
| | | Mid-term | $X_{2:2}$ | Whether the content involves for 3–5 years, Yes for 1, No for 0. |
| | | Short-term | $X_{2:3}$ | Whether the content involves for 1–3 years, Yes for 1, No for 0. |
| Policy Level | $X_3$ | National | $X_{3:1}$ | Whether the main issuing is a national authority, Yes for 1, No for 0. |
| | | Provincial | $X_{3:2}$ | Whether the main issuing is a provincial authority, Yes for 1, No for 0. |
| | | Prefecture | $X_{3:3}$ | Whether the main issuing is a prefecture authority, Yes for 1, No for 0. |
| Policy Field | $X_4$ | Economics | $X_{4:1}$ | Whether the content involves the economic field, Yes for 1, No for 0. |
| | | Society | $X_{4:2}$ | Whether social services are involved, Yes for 1, No for 0. |
| | | Technology | $X_{4:3}$ | Whether the technical level is included, Yes for 1, No for 0. |
| | | Politics | $X_{4:4}$ | Whether it involves the political field, Yes for 1, No for 0. |
| | | Institution | $X_{4:5}$ | Whether it involves the system field, Yes for 1, No for 0. |
| | | Environment | $X_{4:6}$ | Whether the environmental aspect is included, Yes for 1, No for 0. |
| Policy Guarantee | $X_5$ | Technical Support | $X_{5:1}$ | Whether technical support is involved, Yes for 1, No for 0. |
| | | Capital Investment | $X_{5:2}$ | Whether capital investment is involved, Yes for 1, No for 0. |
| | | Industry Standards | $X_{5:3}$ | Whether industry standards are included, Yes for 1, No for 0. |
| | | Infrastructure | $X_{5:4}$ | Whether infrastructure construction is involved, Yes for 1, No for 0. |
| | | Talent Development | $X_{5:5}$ | Whether talent construction is involved, Yes for 1, No for 0. |
| | | Legal Safeguards | $X_{5:6}$ | Whether legal services are involved, Yes for 1, No for 0. |
| | | International Co-operation | $X_{5:7}$ | Whether international cooperation is involved, Yes for 1, No for 0. |
| | | Pilot Promotion | $X_{5:8}$ | Whether pilot promotion is included, Yes for 1, No for 0. |
| | | Publicity and Education | $X_{5:9}$ | Whether publicity and education are included, Yes for 1, No for 0. |
| | | Regulatory Assessment | $X_{5:10}$ | Whether regulatory assessment is involved, Yes for 1, No for 0. |
| | | Public Service | $X_{5:11}$ | Whether public services are involved, Yes for 1, No for 0. |
| Policy Audience | $X_6$ | Government | $X_{6:1}$ | Whether the target group is the government, Yes for 1, No for 0. |
| | | Enterprise | $X_{6:2}$ | Whether the target group is the enterprise, Yes for 1, No for 0. |
| | | Public | $X_{6:3}$ | Whether the target group is the public, Yes for 1, No for 0. |
| | | Universities and Research Institutes | $X_{6:4}$ | Whether the target group is the universities and research institutes, Yes for 1, No for 0. |
| Policy Objectives | $X_7$ | Formation of Factor market | $X_{7:1}$ | Whether the data element market system is formed, Yes for 1, No for 0. |
| | | Industrial Digitization | $X_{7:2}$ | Whether the industrial digital transformation has been driven, Yes for 1, No for 0. |
| | | Digital Industrialization | $X_{7:3}$ | Whether the level of digital industrialization has been improved, Yes for 1, No for 0. |
| | | Equality in Public Services | $X_{7:4}$ | Whether digital public services are inclusive and equal, Yes for 1, No for 0. |
| | | Improvement of Governance System | $X_{7:5}$ | Whether the digital economy governance system is improved, Yes for 1, No for 0. |
| Policy Evaluation | $X_8$ | Sufficient basis | $X_{8:1}$ | Whether the basis is sufficient, Yes for 1, No for 0. |
| | | Clear goals | $X_{8:2}$ | Whether the goals set are clear, Yes for 1, No for 0. |
| | | Scheme science | $X_{8:3}$ | Whether the implemented scheme is scientific, Yes for 1, No for 0. |
| | | Reasonable planning | $X_{8:4}$ | Whether the proposed plan is reasonable, Yes for 1, No for 0. |
| | | Clear rights and responsibilities | $X_{8:5}$ | Whether the rights and responsibilities are clear, Yes for 1, No for 0. |
| Policy Perspective | $X_9$ | Macroscopic | $X_{9:1}$ | Whether it is macro level content, Yes for 1, No for 0. |
| | | Microcosmic | $X_{9:2}$ | Whether it is micro level content, Yes for 1, No for 0. |

*(Continued)*

**Table 3.** (Continued)

| Primary Variables | Code | Secondary Variables | Code | Evaluation Criterion |
|---|---|---|---|---|
| Policy Publicity | $X_{10}$ | — | — | Whether the policy is public, Yes for 1, No for 0. |

Note: The formulation of Policy Type (X1) is based on reference [27]. The formulation of Policy Effectiveness (X2) is based on reference [28]. The formulation of Policy Level (X3) is based on reference [29]. The formulation of Policy Field (X4) is based on reference [30]. The formulation of Policy Guarantee (X5) is based on references [28, 29], and text mining results. The formulation of Policy Audience (X6) is based on references [28, 29], and text mining results. The formulation of Policy Objectives (X7) is based on references [28, 29], and text mining results. The formulation of Policy Evaluation (X8) is based on reference [30]. The formulation of Policy Perspective (X9) is based on reference [26].

are recorded as $P_1$, $P_2$, $P_3$, $P_4$, $P_5$, $P_6$, $P_7$ and $P_8$ respectively. There are 3 national policies and 5 provincial policies. The specific results are shown in Table 5.

## Calculation of PMC index

The calculation of the PMC index model revolves around four aspects [37]. First, inputting the primary and secondary variables from the previous text into a multi-input-output table according to Formula (1). Second, sequentially assigning values to the 45 secondary variables in the digital economic policy multi-input-output table using Formula (2). The values of the secondary variables follow a [0,1] distribution, with a value of 1 assigned if they meet the evaluation criteria and 0 if they do not. Third, calculating the values of the primary variables based on Formula (3). After assigning values to the secondary variables that follow a [0,1] distribution in the digital economic policy PMC index model, the sum of the secondary variable scores is obtained. Then, dividing the sum by the number of secondary variables contained in the respective primary variable yields the arithmetic mean, representing the value of that primary variable. Finally, using Formula (4) to sum the values of each primary variable in the digital economic policy PMC index model, the overall PMC index for each digital economic policy is obtained. The detailed calculation formulas are as follows:

$$X \sim N[0, 1] \tag{1}$$

$$X = \{XR : [0 \sim 1]\} \tag{2}$$

**Table 4. The table of digital economy policy multi-input-output.**

| | $X_1$ | | | | | | $X_2$ | | | |
|---|---|---|---|---|---|---|---|---|---|---|
| $X_{1:1}$ | $X_{1:2}$ | $X_{1:3}$ | $X_{1:4}$ | $X_{1:5}$ | | | $X_{2:1}$ | $X_{2:2}$ | $X_{2:3}$ | |
| | $X_3$ | | | | | $X_4$ | | | | |
| | $X_{3:1}$ | $X_{3:2}$ | $X_{3:3}$ | | $X_{4:1}$ | $X_{4:2}$ | $X_{4:3}$ | $X_{4:4}$ | $X_{4:5}$ | $X_{4:6}$ |
| | | | | | $X_5$ | | | | | |
| $X_{5:1}$ | $X_{5:2}$ | $X_{5:3}$ | $X_{5:4}$ | $X_{5:5}$ | $X_{5:6}$ | $X_{5:7}$ | $X_{5:8}$ | $X_{5:9}$ | $X_{5:10}$ | $X_{5:11}$ |
| | $X_6$ | | | | | $X_7$ | | | | |
| | $X_{6:1}$ | $X_{6:2}$ | $X_{6:3}$ | $X_{6:4}$ | | $X_{7:1}$ | $X_{7:2}$ | $X_{7:3}$ | $X_{7:4}$ | $X_{7:5}$ |
| | $X_8$ | | | | | | $X_9$ | | | $X_{10}$ |
| $X_{8:1}$ | $X_{8:2}$ | $X_{8:3}$ | $X_{8:4}$ | $X_{8:5}$ | | | $X_{9:1}$ | $X_{9:1}$ | | $X_{10}$ |

**Table 5. Policy text to be evaluated for digital economy.**

| Policy Code | Policy Name | Issuing department | Date issued |
|---|---|---|---|
| $P_1$ | The Fourteenth Five Year Plan for Digital Economy Development | The State Council | 12-12-2021 |
| $P_2$ | Guidelines for Foreign Investment and Cooperation in Digital Economy | Ministry of Commerce | 07-20-2021 |
| $P_3$ | Guiding Opinions on Developing Digital Economy, Stabilizing and Expanding Employment | National Development and Reform Commission | 09-18-2018 |
| $P_4$ | Promotion Regulations of Digital Economy of Jiangsu Province | Jiangsu Province | 05-31-2022 |
| $P_5$ | Promotion Regulations of Digital Economy of Hebei Province | Hebei Province | 05-27-2022 |
| $P_6$ | Promotion Regulations of Digital Economy of Henan Province | Henan Province | 12-28-2021 |
| $P_7$ | Promotion Regulations of Digital Economy of Guangdong Province | Guangdong Province | 07-30-2021 |
| $P_8$ | Promotion Regulations of Digital Economy of Zhejiang Province | Zhejiang Province | 12-24-2020 |

$$X_t \left( \sum_{j=1}^{n} \frac{X_{tj}}{T\left(X_{tj}\right)} \right), t = 1, 2, 3, 4, 5, 6, 7, 8, 9, 10, \ldots\ldots, \infty \tag{3}$$

As shown in the formula, $t$ is the primary variable, $j$ is the secondary variable and $T(X_{tj})$ is the number of secondary indicators under the primary indicator.

$$PMC = \left\{ \begin{array}{l} X_1 \left( \sum_{i=1}^{5} \frac{X_{1i}}{5} \right) + \quad X_2 \left( \sum_{j=1}^{3} \frac{X_{2j}}{3} \right) + \quad X_3 \left( \sum_{k=1}^{3} \frac{X_{3k}}{3} \right) + \\ X_4 \left( \sum_{l=1}^{6} \frac{X_{4l}}{6} \right) + \quad X_5 \left( \sum_{m=1}^{11} \frac{X_{5m}}{11} \right) + \quad X_6 \left( \sum_{n=1}^{4} \frac{X_{6n}}{4} \right) + \\ X_7 \left( \sum_{o=1}^{5} \frac{X_{7o}}{5} \right) + \quad X_8 \left( \sum_{p=1}^{5} \frac{X_{8p}}{5} \right) + \quad X_9 \left( \sum_{r=1}^{2} \frac{X_{9r}}{2} \right) + X_{10} \end{array} \right\} \tag{4}$$

According to the formula (1)-(4), the calculation results are brought into Table 4. Therefore, the table of multi-input-output of digital economic policies of China is finally obtained, as shown in Table 6. Simultaneously, referring to Ruiz's classification standards for policies, such as 9–10 (perfect), 7–8.99 (excellent), 5–6.99 (acceptable), and 0–4.99 (defective). Finally, the PMC index and evaluation grade of digital economic policy are determined, as shown in Table 7.

**Surface plot of PMC.** The surface plot can show the quantitative results more intuitively and the differences between various policies. The fluctuation degree of the surface plot can be used to judge the gaps of the policies. The smaller the fluctuation degree is, the more reasonable the internal structure of the policy is, and the more detailed the policy is.

The premise of constructing surface plot of PMC is to calculate the matrix of PMC. The matrix of PMC is a 3×3 matrix, composed of 9 primary variables. Currently, there are 10 primary variables. However, the $X_{10}$ of primary variable does not have any secondary variable, and its policy scores are all 1. Therefore, the $X_{10}$ of the primary variable is eliminated under the premise of considering matrix symmetry. Finally, the 3×3 matrix is constructed by 9 primary variables. It can more intuitively show the consistency and rationality within the policy.

**Table 6. The table of multi input and output of digital economic policies.**

| Primary Variables | Secondary Variables | $P_1$ | $P_2$ | $P_3$ | $P_4$ | $P_5$ | $P_6$ | $P_7$ | $P_8$ |
|---|---|---|---|---|---|---|---|---|---|
| $X_1$ | $X_{1:1}$ | 1 | 0 | 1 | 0 | 0 | 0 | 0 | 0 |
| | $X_{1:2}$ | 1 | 1 | 1 | 1 | 1 | 1 | 1 | 1 |
| | $X_{1:3}$ | 1 | 1 | 1 | 1 | 1 | 1 | 1 | 1 |
| | $X_{1:4}$ | 1 | 1 | 1 | 1 | 1 | 1 | 1 | 1 |
| | $X_{1:5}$ | 1 | 1 | 1 | 1 | 1 | 1 | 1 | 1 |
| $X_2$ | $X_{2:1}$ | 1 | 0 | 1 | 0 | 0 | 0 | 0 | 0 |
| | $X_{2:2}$ | 0 | 0 | 0 | 1 | 1 | 1 | 1 | 1 |
| | $X_{2:3}$ | 0 | 1 | 0 | 0 | 0 | 0 | 0 | 0 |
| $X_3$ | $X_{3:1}$ | 1 | 1 | 1 | 0 | 0 | 0 | 0 | 0 |
| | $X_{3:2}$ | 0 | 0 | 0 | 1 | 1 | 1 | 1 | 0 |
| | $X_{3:3}$ | 0 | 0 | 0 | 0 | 0 | 0 | 0 | 0 |
| $X_4$ | $X_{4:1}$ | 1 | 1 | 1 | 1 | 1 | 1 | 1 | 1 |
| | $X_{4:2}$ | 1 | 0 | 1 | 1 | 1 | 1 | 1 | 1 |
| | $X_{4:3}$ | 1 | 1 | 1 | 1 | 1 | 1 | 1 | 1 |
| | $X_{4:4}$ | 0 | 0 | 0 | 0 | 0 | 0 | 0 | 0 |
| | $X_{4:5}$ | 1 | 1 | 1 | 1 | 1 | 1 | 0 | 0 |
| | $X_{4:6}$ | 0 | 0 | 0 | 1 | 1 | 1 | 1 | 1 |
| $X_5$ | $X_{5:1}$ | 1 | 1 | 1 | 1 | 1 | 1 | 1 | 1 |
| | $X_{5:2}$ | 1 | 0 | 0 | 1 | 1 | 1 | 1 | 1 |
| | $X_{5:3}$ | 1 | 1 | 1 | 1 | 1 | 1 | 1 | 1 |
| | $X_{5:4}$ | 1 | 1 | 1 | 1 | 1 | 1 | 1 | 1 |
| | $X_{5:5}$ | 1 | 1 | 1 | 1 | 1 | 1 | 1 | 1 |
| | $X_{5:6}$ | 0 | 0 | 1 | 0 | 0 | 0 | 0 | 0 |
| | $X_{5:7}$ | 1 | 1 | 0 | 0 | 1 | 1 | 1 | 1 |
| | $X_{5:8}$ | 1 | 0 | 1 | 1 | 1 | 0 | 0 | 1 |
| | $X_{5:9}$ | 1 | 0 | 0 | 1 | 1 | 1 | 1 | 1 |
| | $X_{5:10}$ | 1 | 1 | 1 | 1 | 1 | 1 | 1 | 1 |
| | $X_{5:11}$ | 1 | 1 | 1 | 1 | 1 | 1 | 1 | 1 |
| $X_6$ | $X_{6:1}$ | 1 | 1 | 1 | 1 | 1 | 1 | 1 | 1 |
| | $X_{6:2}$ | 1 | 1 | 1 | 1 | 1 | 1 | 1 | 1 |
| | $X_{6:3}$ | 1 | 0 | 1 | 1 | 1 | 1 | 1 | 1 |
| | $X_{6:4}$ | 1 | 1 | 1 | 1 | 1 | 1 | 1 | 1 |
| $X_7$ | $X_{7:1}$ | 1 | 1 | 1 | 1 | 1 | 1 | 0 | 1 |
| | $X_{7:2}$ | 1 | 1 | 1 | 1 | 1 | 1 | 1 | 1 |
| | $X_{7:3}$ | 1 | 1 | 1 | 1 | 1 | 1 | 1 | 1 |
| | $X_{7:4}$ | 1 | 1 | 1 | 1 | 0 | 0 | 0 | 1 |
| | $X_{7:5}$ | 1 | 0 | 0 | 1 | 1 | 1 | 0 | 1 |
| $X_8$ | $X_{8:1}$ | 1 | 1 | 1 | 1 | 1 | 1 | 1 | 1 |
| | $X_{8:2}$ | 1 | 1 | 1 | 1 | 1 | 1 | 1 | 1 |
| | $X_{8:3}$ | 1 | 1 | 1 | 1 | 1 | 1 | 1 | 1 |
| | $X_{8:4}$ | 1 | 1 | 1 | 1 | 1 | 1 | 1 | 1 |
| | $X_{8:5}$ | 1 | 0 | 1 | 1 | 1 | 1 | 1 | 1 |
| $X_9$ | $X_{9:1}$ | 1 | 1 | 1 | 0 | 0 | 0 | 0 | 0 |
| | $X_{9:2}$ | 0 | 0 | 0 | 1 | 1 | 1 | 1 | 1 |
| $X_{10}$ | $X_{10}$ | 1 | 1 | 1 | 1 | 1 | 1 | 1 | 1 |

**Table 7. The PMC index and evaluation grade of digital economic policy.**

| Primary Variable | $P_1$ | $P_2$ | $P_3$ | $P_4$ | $P_5$ | $P_6$ | $P_7$ | $P_8$ | Mean Value |
|---|---|---|---|---|---|---|---|---|---|
| $X_1$ | 1.00 | 0.80 | 1.00 | 0.80 | 0.80 | 0.80 | 0.80 | 0.80 | 0.85 |
| $X_2$ | 0.33 | 0.33 | 0.33 | 0.33 | 0.33 | 0.33 | 0.33 | 0.33 | 0.33 |
| $X_3$ | 0.33 | 0.33 | 0.33 | 0.33 | 0.33 | 0.33 | 0.33 | 0.33 | 0.33 |
| $X_4$ | 0.67 | 0.50 | 0.67 | 0.83 | 0.83 | 0.83 | 0.67 | 0.67 | 0.71 |
| $X_5$ | 0.91 | 0.64 | 0.73 | 0.82 | 0.91 | 0.82 | 0.82 | 0.91 | 0.82 |
| $X_6$ | 1.00 | 0.75 | 1.00 | 1.00 | 1.00 | 1.00 | 1.00 | 1.00 | 0.97 |
| $X_7$ | 1.00 | 0.80 | 0.80 | 1.00 | 0.80 | 0.80 | 0.40 | 1.00 | 0.83 |
| $X_8$ | 1.00 | 0.80 | 1.00 | 1.00 | 1.00 | 1.00 | 1.00 | 1.00 | 0.98 |
| $X_9$ | 0.50 | 0.50 | 0.50 | 0.50 | 0.50 | 0.50 | 0.50 | 0.50 | 0.50 |
| $X_{10}$ | 1.00 | 1.00 | 1.00 | 1.00 | 1.00 | 1.00 | 1.00 | 1.00 | 1.00 |
| PMC Index | 7.74 | 6.45 | 7.36 | 7.61 | 7.50 | 7.41 | 6.85 | 7.54 | 7.31 |
| Ranking | 1 | 8 | 6 | 2 | 4 | 5 | 7 | 3 | — |
| Grade | E | A | E | E | E | E | A | E | — |

Note: E stands for Excellent, A stands for Acceptable.

The calculation of PMC surface is shown in formula (5).

$$\text{PMC } surface = \begin{pmatrix} X_1 & X_2 & X_3 \\ X_4 & X_5 & X_6 \\ X_7 & X_8 & X_9 \end{pmatrix} \tag{5}$$

The PMC surface of digital economy policy of China is shown in Fig 1. The X axis corresponds to 1, 2 and 3 in the Fig 1, and the Y axis corresponds to series 1, 2 and 3. The Z axis represents the score of the PMC index of the policy to be evaluated. Different color blocks represent different scores of the primary variables. The convex part of the surface plot indicates that there is a high score. The concave part of the surface plot indicates that there is a low score. The advantages and disadvantages of a certain policy can be seen from the comparison among the eight policies. The PMC surfaces of the policies of $P_1$, $P_3$, $P_4$, $P_5$, $P_6$ and $P_8$ have a certain degree of concavity and convexity, indicating that the internal consistency is relatively high and the structure is relatively reasonable. The PMC surfaces of the policies of $P_7$ and $P_2$ have an obvious fluctuation trend, indicating that the internal consistency is low, and the policy is not detailed enough, and the overall score is low.

**Analysis of quantitative results.** Among the 8 selected digital economic policies, there are 6 excellent policies (2 national policies and 4 provincial policies) and 2 acceptable policies (1 national policy and 1 provincial policy). Furthermore, there is no perfect policy or bad policy. The specific ranking of these policies is as follows $P_1 > P_4 > P_8 > P_5 > P_6 > P_3 > P_7 > P_2$. On the whole, the digital economy policy of China is scientific and reasonable. The central government and the local governments have a strong sense of coordination in policy formulation. It has effectively promoted the preliminary establishment of the data resource element market and the rapid development of the digital economy industry in China. However, it is worth noting that the lack of perfect policies indicates that the quality of current digital economic policies still has some room for improvement.

The radar map is made according to the average value of 9 primary variables in the 8 policies, which can intuitively and clearly show the shortcomings of digital economic policies of China, as shown in Fig 2. It is an important aspect that needs attention in the process of formulating digital economy policies in the future.

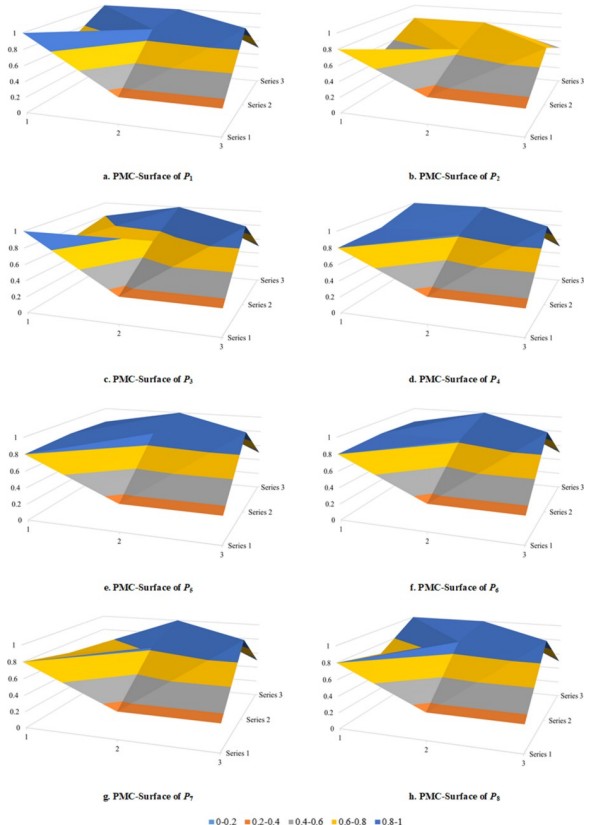

**Fig 1. PMC-surface of digital economic policy.**

The average value of policy type $X_1$ is 0.85, which shows that digital economy policy of China has a strong role in supervision, suggestion, description and guidance. The average value of policy effectiveness $X_2$ is 0.33, which shows there is no effective connection among

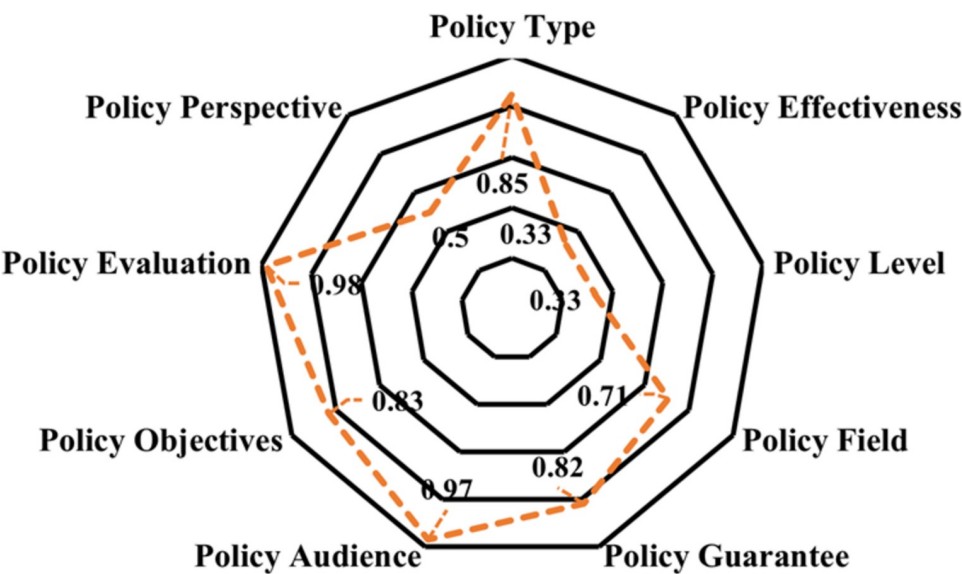

**Fig 2. Radar map of digital economy policy.**

long-term, medium-term and short-term in the process of formulating digital economy policies. Too much attention is paid to the development of medium and short term policies, ignoring the long term implementation of policies. The average value of policy level $X_3$ is 0.33. It shows that the issuing units of relevant policies are mainly independent departments and lack of joint issuing organs. The average value of policy areas $X_4$ is 0.71. It shows that the current digital economy policy covers a wide range of fields and involves a deeper level. The average value of policy guarantee $X_5$ is 0.82. It shows that China has used many kinds of security methods to formulate digital economy policies. The average value of policy audience $X_6$ is 0.97. It shows that the digital economy policy involves a wide audience. The average value of policy objectives $X_7$ is 0.83, which shows that the digital economy policy has clear objectives. The average value of policy evaluation $X_8$ is 0.98. It fully shows that the formulation of relevant policies is based on sufficient, reasonable planning and scientific scheme. The average value of policy perspective $X_9$ is 0.5. It shows that the macro and micro levels are not fully involved in the formulation of relevant policies, and the combination of macro and micro is low.

The PMC index of the fourteenth five year plan for digital economy is 7.74. The rank of $P_1$ is No. 1, and its ranking is excellent. Only $X_4$ of the primary variables is below the mean value. The planning outline of digital economy development of China has fully played a leading role, reflected the confidence and the ability of accelerating the cultivation and development of the digital economy, promoting industrial reform, and enhancing comprehensive competitiveness. Therefore, if the policy is to be improved, the indicator of $X_4$ should be given priority.

The PMC index of the promotion regulations of digital economy of Jiangsu Province is 7.61. The rank of $P_4$ is No. 2, and its ranking is excellent. Only $X_1$ of the primary variables is below the mean value. The rank of this policy is No. 1 among the five provincial policies selected, and its score exceeds two national policies. It means that the policy makers consider comprehensively and reasonably when designing policies. Resource elements, digital industry, public service and governance system are considered comprehensively. However, the $X_1$ of the primary variables does not involve prediction, and it becomes the optimization object in future.

The PMC index of the promotion regulations of digital economy of Zhejiang Province is 7.54. The rank of $P_8$ is No. 3, and its ranking is excellent. Only $X_1$ and $X_4$ of the primary variables are below the mean value. Different from the $P_1$ of the national policy, the $P_8$ of the provincial policy significantly weakens the prediction of the digital economy industry, and more considers the content of the micro level. Therefore, the optimization path is from $X_1$ to $X_4$ in future.

The PMC index of the promotion regulations of digital economy of Hebei Province is 7.50. The rank of $P_5$ is No. 4, and its ranking is excellent. Only $X_1$ and $X_7$ of the primary variables are below the mean value. The full and detailed of policy guarantee reflects the confidence and determination. However, too many policy objectives focus on industrial digitization and digital industrialization, which does not reflect equal and inclusive public services. Therefore, the optimization path is from $X_1$ to $X_7$ in future.

The PMC index of the promotion regulations of digital economy of Henan Province is 7.41. The rank of $P_6$ is No. 5, and its ranking is excellent. Only $X_1$ and $X_7$ of the primary variables are below the mean value. The same as $P_5$, the policy types do not include predictability, and the policy objectives do not cover digital public services. For Henan, the supporting role of public services and social governance should be strengthened when making policies. Furthermore, the service system should be optimized to promote the development of digital economy in the region and build a strong province of digital economy. Therefore, the optimization path is from $X_1$ to $X_7$ in future.

The PMC index of guiding opinions on developing digital economy, stabilizing and expanding employment is 7.36. The rank of $P_3$ is No. 6, and its ranking is excellent. $X_4$, $X_5$, and $X_7$ of the primary variables are below the mean value. The main goal of policy formulation is to achieve higher quality and full employment, and constantly expand employment innovation space. However, there are few policy guarantees, and the policy objectives fail to reflect the improvement of governance system of digital economy. Therefore, the optimization path is from $X_5$ to $X_4$ to $X_7$ in future.

The PMC index of the promotion regulations of digital economy of Guangdong Province is 6.85. The rank of $P_7$ is No. 7, and its ranking is acceptable. $X_1$, $X_4$, and $X_7$ of the primary variables are below the mean value. The difference between $X_7$ and the mean value is the largest. It indicates that the policy objective setting needs to be strengthened. It is necessary to further optimize policies to promote high-quality economic development in combination with the documents and regulations of national strategic and the regional industry situation. Therefore, the optimization path is from $X_7$ to $X_4$ to $X_1$ in future.

The PMC index of guidelines for foreign investment and cooperation in digital economy is 6.45. The rank of $P_2$ is No. 8, and its ranking is acceptable. Six of the primary variables are lower than the mean value such as $X_1$, $X_4$, $X_5$, $X_6$, $X_7$, and $X_8$. This policy is mainly guided by foreign investment and cooperation, and its content is not as comprehensive and detailed as other policies to be evaluated. The difference between $X_6$ and the mean value is the largest, and the difference between $X_7$ and the mean value is the smallest. Therefore, the optimization path is $X_6$ - $X_4$ - $X_8$ - $X_5$ - $X_1$ - $X_7$ in future.

## Conclusions and suggestions

### Conclusions

This paper conduct content analysis and text mining on 37 digital economic policies. Then, the PMC index model of digital economic policy of China was constructed. Therefore, the empirical analysis was conducted on the eight selected representative policies. The main research conclusions are as follows.

Firstly, the overall design of digital economy policy of China is more reasonable and comprehensive. The average of PMC index of the eight evaluated policy texts is 7.31. There are the 6 policies of $P_1$, $P_3$, $P_4$, $P_5$, $P_6$, and $P_8$ evaluated as excellent. Among them, the provincial policies account for a large proportion. There are the 2 policies of $P_2$ and $P_7$ evaluated as acceptable. The provincial policies account for half. It fully shows that China has fully considered the current situation and future development trend of the industry when formulating digital economy policies, and has promoted the construction of development pattern of digital economy under the background of industrial reform.

Secondly, according to the multi-input-output of digital economic policies, the digital economic policy of China still has a lot of room for improvement. The policy type of $X_1$ mainly focuses on supervision, advice, description and guidance, and lacks prediction. Only the national policies of $P_1$ and $P_3$ involve prediction, while the provincial policies of $P_4$, $P_5$, $P_6$, $P_7$, and $P_8$ do not involve prediction. Although the policy effectiveness of $X_2$ involves the policies of long-term, mid-term and short-term, but the single policy does not well reflect the policies combination of long-term, mid-term and short-term. In fact, perfect policies should effectively combine the long-term planning with the short-term goals. The overall score of the policy guarantee of $X_5$ is high. However, there is a serious lack of legal protection. Among the 8 policies to be evaluated, only $P_3$ involves policy guarantee. The policy objectives of $X_7$ does not highlight the function of public service especially for the provincial policies. Therefore, it is not conducive to the further improvement of the overall service level.

## Suggestions

Considering the uniqueness of the digital economy and in conjunction with the primary research findings, the subsequent recommendations are proposed.

Firstly, China's digital economic policies predominantly emphasize supervision, suggestions, descriptions, and guidance, while lacking predictive aspects. A scientifically grounded and rational prediction is instrumental in cultivating the digital economy's factor market and directing standardized development. Policies encompassing predictive mechanisms can facilitate broader societal involvement and stimulate industrial growth at various levels. Moreover, they can enhance the efficacy of governmental departments, augmenting their functional capabilities and executive authority, thus refining decision-making efficiency.

Secondly, the selected policies primarily center on mid-term and long-term leading strategies. Long-term policies exhibit a strategic, forward-looking nature that fosters a favorable business climate for the digital economy sector. However, these policies hold a broader guiding influence while offering limited guidance in specific domains. Consequently, the implementation of specialized policies can enhance the standardization of the digital economy sector. Simultaneously, it is imperative to establish effective linkages and dynamic synergies among long-term, mid-term, and short-term policies, instead of solely pursuing rapid short-term policy outcomes.

Thirdly, policy guarantee is a means and measure taken by the government to achieve a certain set goal. Good policy guarantees can successfully protect the digital economy. At present, the legislation in the digital field should be expedited, and the digital governance framework should be optimized. Sound laws are conducive to optimizing the business environment and protecting the legitimate rights and interests of consumers. Attracting enterprises and talents from all over the country could promote the development of technology and the accumulation of capital, and improve the overall social welfare level.

Finally, pay attention to the role of stakeholders in the digital economy. In the era of digital economy, stakeholders have already extended to employees, users, and partners. Stakeholders should not only include shareholders, but also employees in the future. We should not only focus on shareholders, but also on employees. Only with excellent employees can we create greater value for the enterprise and achieve better development. The digital economy is built on the basis of data, and enterprises earn more profits by collecting user data. Enterprises should be regulated while using data to earn more profits. Regulators need to play a more important role in protecting user privacy and data usage. Protecting privacy requires stricter regulations to ensure it. The government is facing a new market composed of an increasing number of platforms, which poses an unprecedented challenge to its market governance. On the basis of a scientific understanding of the new economic form of the digital economy and a rational analysis of the two growth paths of the digital economy, the government should redesign the content of government management functions in the digital economy and further play a positive role.

## Research limitations and future trends

It is important to note that this study primarily quantitatively evaluated China's national and provincial-level digital economic policies, excluding policies at the municipal and county levels. Due to variations in the level of digital industry development among different municipalities, a unified calculation standard has not yet been established. Therefore, future research should aim to expand the sample size and diversify the selection of subjects. Additionally, a comprehensive cross-analysis of policy content can be conducted from different dimensions by integrating policy tool theories. This approach can offer decision-makers objective and

feasible solutions. Additionally, in terms of policy evaluation methods, since this study exclusively employed the PMC index model, the results are relatively singular. In the future, a more comprehensive approach should be considered, incorporating various evaluation methods rather than relying solely on a single approach.

Building upon the research findings of this paper and considering the current development of China's digital economic industry, future studies can be directed towards the following aspects:

On one hand, it is essential to accelerate the integration of the digital economy with the real economy. As the digital economy and its systems become increasingly sophisticated, policies need to be appropriately adjusted, encompassing goals, positioning, and tools. Focusing on digital industrialization and industry digitization, policies should provide broader application scenarios for digital economic development, amplifying and multiplying the impact of digital technologies on economic growth.

On the other hand, emphasis should be placed on innovation-driven development. Presently, there are noticeable disparities in the development of the digital economy across the eastern, central, and western regions of China. Innovation-driven approaches can bridge these regional gaps, serving as a crucial lever for regional coordination. This strategy enables the utilization of developed digital economic regions' driving roles and innovation advantages, enhances regional collaborative innovation capabilities, and promotes high-quality development of the digital economic industry.

Lastly, in order to achieve the policy objectives of digital industrialization and industry digitization, internal demand needs to be constantly improved and external demand needs to be fully utilized. The development of digital economy is long-term and systematic. Therefore, it is needed to accelerate the establishment of data element market, promote the equality and benefits of public services, improve the governance system of the digital economy, eliminate the digital divide and lay a solid foundation for building a good modern market system of digital economy.

## Supporting information

**S1 Appendix. 37 original policy docuements.**
(DOCX)

**S2 Appendix. List of policy names.**
(XLSX)

## Author Contributions

**Conceptualization:** Tianzun Wang.

**Data curation:** Tianzun Wang, Xiaoyi Fu, Guo Li.

**Formal analysis:** Tianzun Wang.

**Methodology:** Shuai Hong, Tianzun Wang.

**Supervision:** Shuai Hong.

**Validation:** Shuai Hong.

**Visualization:** Tianzun Wang.

**Writing – original draft:** Tianzun Wang.

**Writing – review & editing:** Shuai Hong, Tianzun Wang.

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
