## [Decision Letter · Decision Letter 0]

9 May 2023

PONE-D-23-07522Research on Quantitative Evaluation of Digital Economy Policy in China Based on the PMC Index ModelPLOS ONE

Dear Dr. WANG,

Thank you for submitting your manuscript to PLOS ONE. After careful consideration, we feel that it has merit but does not fully meet PLOS ONE’s publication criteria as it currently stands. Therefore, we invite you to submit a revised version of the manuscript that addresses the points raised during the review process.

We look forward to receiving your revised manuscript.

Kind regards,

Jing Cheng

Academic Editor

PLOS ONE

Journal Requirements:

"33602"

Reviewers' comments:

Reviewer's Responses to Questions

**Comments to the Author**

1. Is the manuscript technically sound, and do the data support the conclusions?

Reviewer #1: Partly

Reviewer #2: Yes

Reviewer #3: Yes

2. Has the statistical analysis been performed appropriately and rigorously? 

Reviewer #1: No

Reviewer #2: N/A

Reviewer #3: Yes

3. Have the authors made all data underlying the findings in their manuscript fully available?

Reviewer #1: No

Reviewer #2: No

Reviewer #3: Yes

4. Is the manuscript presented in an intelligible fashion and written in standard English?

Reviewer #1: No

Reviewer #2: Yes

Reviewer #3: Yes

5. Review Comments to the Author

Reviewer #1: Dear Editor,

I am pleased to read the manuscript (PONE-D-23-07522) entitled "Research on Quantitative Evaluation of Digital Economy Policy in China Based on the PMC Index Model" The topic is very interesting, and the approach used is justified in the literature. The paper is well written. In my opinion, it also represents an unexplored, interesting topic to be investigated. This paper is well presented and can be accepted after addressing these major recommendations.

1. In order to establish the significance and value of the study, it is necessary to provide a comprehensive rationale for the research, which emphasizes its relevance and unique contributions to the current scholarly discourse. This will strengthen the study's originality and scholarly impact. Following are useful suggested studies to get the benefit to update this part:https://doi.org/10.1007/s11356-022-19718-6;https://doi.org/10.3390/su14031054;
https://doi.org/10.1007/s11356-021-17438-x;https://doi.org/10.1108/FS-02-2021-0053;
https://doi.org/10.1007/s11356-022-209227;https://doi.org/10.1177/21582440211061554;DOI: 10.26677/TR1010.2019.145; https://doi.org/10.3389/fenvs.2022.967418;https://doi.org/10.1177/0958305X221134113

2. English check/editing: I found major typos/ grammatical, this study was presented with depraved English language usage.

3. Recheck throughout the paper and ensure that all abbreviations are defined the first time they are used.

4. The paper clearly lacks the sense of paragraphing; some paragraphs are too shorting and consist of only few lines. Kindly rearrange your paragraphs and provide arguments about one main theme in each paragraph.

5. The Results and described with reference to the references of the parameters that influence the results of the research will make it more original. The purpose of the study and the reasons clearly set forth.

6. Please increase the pixel size of all Figures.

7. The conclusion is sufficiently discussed with some useful policy implications. However, limitations and future directions of research should have a reasonably sized paragraph instead of just a few lines written half-heartedly at the end of the conclusion.

8. Follow the right style of citation throughout the manuscript by checking the guidelines of (PLOS ONE) journal or any previously published paper in the journal.

Reviewer #2: This manuscript studied quantitative evaluation of digital economy policy in China based on the PMC index model. Although the topic is interesting, the innovation and contribution is not high. In addition, there are some shortcomings in this manuscript. Firslty, variable setting of PMC model of digital economy policy is not clearly. Please illustrate in detail. Secondly, how to testify the efficency of the PMC model? Can you compare with other models? Finally, the corresponding countermeasures and suggestions should be more concrete and can be executed.

Reviewer #3: First of al, I congratulates to the authors for making a significant empirical contribution in economic literature.The author implemented significant contribution in order to have their work published in the PLOS One journal.

6. PLOS authors have the option to publish the peer review history of their article (what does this mean?). If published, this will include your full peer review and any attached files.

Reviewer #1: **Yes: **KASHIF ABBASS

Reviewer #2: No

Reviewer #3: **Yes: **Dr. Shabbir Ahmed Department of economics Government islamia graduate college kasur pakistan

---

## [Author Response · Author response to Decision Letter 0]

12 Aug 2023

Dear Editor and Reviewers.

Thank you for offering us an opportunity to improve the quality of our submitted manuscript (PONE-D-23-07522R1). We appreciated very much the reviewers’ constructive and insightful comments. In this revision, we have addressed all of these comments. We hope the revised manuscript has now met the publication standard of your journal.

We highlighted all the revisions in yellow colour.

On the next pages, our point-to-point responses to the queries raised by the reviewers are listed.

Journal Requirements:

Response: Thanks for your advice. We have performed a full-text revision according to the style requirements of the PLOS ONE journal.

Response: Thanks for your suggestions, we have provided the correct funding information: The National Social Science Foundation of China, 21BJL073, A/Prof. Xuebin Tian.

"33602"

Response: The funders had no role, and we have stated this in the manuscript: The funder had no role in study design, data collection and analysis, decision to publish, or preparation of the manuscript.

Response: We have uploaded the relevant data to the manuscript and made it editable in the revised cover letter. Supporting information: S1 Appendix. 37 original policy documents. (DOCX), S2 Appendix. List of policy names (XLSX). Data availability statement: All relevant data are within the paper and its Supporting Information fields. 

DOIs: https://doi.org/10.5061/dryad.0rxwdbs5s

Response: Thanks for your suggestion, We have registered an ORCID ID and verified it in the journal's submission system.

Reviewer #1:

1. In order to establish the significance and value of the study, it is necessary to provide a comprehensive rationale for the research, which emphasizes its relevance and unique contributions to the current scholarly discourse. This will strengthen the study's originality and scholarly impact. Following are useful suggested studies to get the benefit to update this part:https://doi.org/10.1007/s11356-022-19718-6;https://doi.org/10.3390/su14031054;
https://doi.org/10.1007/s11356-021-17438-x;https://doi.org/10.1108/FS-02-2021-0053;
https://doi.org/10.1007/s11356-022-209227;https://doi.org/10.1177/21582440211061554;DOI: 10.26677/TR1010.2019.145; https://doi.org/10.3389/fenvs.2022.967418;https://doi.org/10.1177/0958305X221134113

Response: We have optimized and revised the theoretical foundation based on the references provided by the peer reviewers and aligned it with the direction of our research. The specific changes are as follows:

The theory of public policy evaluation is a theoretical framework for assessing policies, aimed at determining their effectiveness, feasibility, sustainability, and fairness. Its scope of application encompasses various domains, including government policies, social policies, and economic policies. On one hand, this theory enhances the scientific and practical aspects of policy formulation by offering a scientific and systematic approach to evaluating policy effects. On the other hand, evaluation outcomes assist policymakers in comprehending the effects of policy implementation, thereby enabling policy adjustments and improvements based on the evaluation results. Existing methods for public policy evaluation each possess distinctive characteristics, providing crucial references for assessments related to digital economic policies. Due to the distinct scopes of application for each method, evaluation outcomes also exhibit slight variations. The grey relational degree model exhibits subjectivity and substantial error in determining optimal values for various indicators. The slow convergence of the BP neural network hampers the precision of policy evaluation. The fuzzy comprehensive evaluation method lacks objectivity in assessing indicators. It's noteworthy that current policy evaluation research excessively emphasizes ex-post evaluation, paying insufficient attention to the policies themselves. In contrast, the PMC (Policy Modeling Consistency) index model showcases a more prominent comprehensive evaluation effect. This method underscores the internal consistency of policies, offering advantages such as diversity assessment, pliability of indicators, and distinguishable gradations. Additionally, serving as a model for evaluating policy texts, it boasts traits like low cost and ease of operation, which mitigate subjectivity during the evaluation process. While assessing the merits and demerits of individual policies, it better reveals policy disparities across different periods or regions through policy comparisons.

Meanwhile, we have also made certain changes to the contributions of this article:

The contributions of this study are as follows:

Quantitative assessment of national and provincial-level digital economic policy texts in China is conducted through the utilization of content analysis and text mining methods. This endeavor aims to gain deeper insights into the ongoing trends within China's digital economic industry. Moreover, it seeks to furnish optimized recommendations for the formulation and refinement of future policies. Simultaneously, from an empirical standpoint, a quantitative evaluation of China's digital economic policies is performed. The outcomes of policy evaluation serve as a reference for standardizing and enhancing the practicality of the future digital economic industry policy evaluation system, thereby facilitating the high-quality development of China's digital economic sector.

2. English check/editing: I found major typos/ grammatical, this study was presented with depraved English language usage.

Response: Thank you for the reviewer's feedback. We have made corrections to all the spelling and grammar errors in the article.

3. Recheck throughout the paper and ensure that all abbreviations are defined the first time they are used.

Response: Thanks for your suggestion. We have provided definitions for terms such as PMC (Policy Modeling Consistency) index model and "PKULaw" (Peking University Legal Information Retrieval System) legal database in their initial instances of use within the article.

4. The paper clearly lacks the sense of paragraphing; some paragraphs are too shorting and consist of only few lines. Kindly rearrange your paragraphs and provide arguments about one main theme in each paragraph.

Response: Thanks for your advice. We have merged the five paragraphs of the article into four paragraphs, while also summarizing and arranging the themes of each paragraph.

5. The Results and described with reference to the references of the parameters that influence the results of the research will make it more original. The purpose of the study and the reasons clearly set forth.

Response: The research objective of this paper is to employ the PMC index model's research methodology to provide an in-depth interpretation of China's digital economic policies. Building upon text mining and content analysis, this study employs a combined quantitative and qualitative approach to thoroughly explore the strengths and weaknesses of China's digital economic policies. Subsequently, it presents corresponding optimization paths and recommendations, aiming to furnish scientific decision-making foundations for expediting the development of China's digital economic market.

6. Please increase the pixel size of all Figures.

Response: We have increased the pixel dimensions of the eight figures in Figure 1.

7. The conclusion is sufficiently discussed with some useful policy implications. However, limitations and future directions of research should have a reasonably sized paragraph instead of just a few lines written half-heartedly at the end of the conclusion.

Response: Thank you for the suggestions from the reviewer. We have summarized the limitations of this study and outlined future research directions.

Research Limitations and Future Trends

It should be noted that this study primarily analyzed national and provincial policies in China and did not select city-level policies within various provinces. Due to variations in the level of digital industry development among different municipalities, a unified calculation standard has not yet been established. Therefore, future research should aim to expand the sample size and diversify the selection of subjects. Additionally, a comprehensive cross-analysis of policy content can be conducted from different dimensions by integrating policy tool theories. This approach can offer decision-makers objective and feasible solutions. Regarding policy quantification assessment, experimenting with various evaluation methods and selecting the optimal option after comprehensive comparisons could be attempted.

Building upon the research findings of this paper and considering the current development of China's digital economic industry, future studies can be directed towards the following aspects:

On one hand, it is essential to accelerate the integration of the digital economy with the real economy. As the digital economy and its systems become increasingly sophisticated, policies need to be appropriately adjusted, encompassing goals, positioning, and tools. Focusing on digital industrialization and industry digitization, policies should provide broader application scenarios for digital economic development, amplifying and multiplying the impact of digital technologies on economic growth.

On the other hand, emphasis should be placed on innovation-driven development. Presently, there are noticeable disparities in the development of the digital economy across the eastern, central, and western regions of China. Innovation-driven approaches can bridge these regional gaps, serving as a crucial lever for regional coordination. This strategy enables the utilization of developed digital economic regions' driving roles and innovation advantages, enhances regional collaborative innovation capabilities, and promotes high-quality development of the digital economic industry.

Lastly, in order to achieve the policy objectives of digital industrialization and industry digitization, internal demand needs to be constantly improved and external demand needs to be fully utilized. The development of digital economy is long-term and systematic. Therefore, it is needed to accelerate the establishment of data element market, promote the equality and benefits of public services, improve the governance system of the digital economy, eliminate the digital divide and lay a solid foundation for building a good modern market system of digital economy.

8. Follow the right style of citation throughout the manuscript by checking the guidelines of (PLOS ONE) journal or any previously published paper in the journal.

Response: Thanks for pointing out the mistake. We have made all the necessary changes to adhere to the reference citation style of the PLOS ONE journal.

References

1. Litvinenko VS. Digital Economy as a Factor in the Technological Development of the Mineral Sector. Natural Resources Research. 2020;29(3):1521-41. doi: 10.1007/s11053-019-09568-4. PubMed PMID: WOS:000492018400001.

2. Meng FS, Zhao Y. How does digital economy affect green total factor productivity at the industry level in China: from a perspective of global value chain. Environmental Science and Pollution Research. 2022;29(52):79497-515. doi: 10.1007/s11356-022-21434-0. PubMed PMID: WOS:000812488200005.

3. Urinovich Kobilov A, Khashimova DP, Mannanova SG, Ogli Abdulakhatov MM. Modern Content and Concept of Digital Economy. International Journal of Multicultural and Multireligious Understanding. 2022;9(2). doi: 10.18415/ijmmu.v9i2.3524.

4. Wang J, Dong K, Dong X, Taghizadeh-Hesary FJEE. Assessing the digital economy and its carbon-mitigation effects: The case of China. 2022;113:106198.

5. Sturgeon TJJGsj. Upgrading strategies for the digital economy. 2021;11(1):34-57.

6. Chen YJCER. Improving market performance in the digital economy. 2020;62:101482.

7. Vasyltsiv T, Lupak R, Kunytska-Iliash M, Shtets TJZNWwP. Trends in state policy with a view to improving structural characteristics of the digital economy. 2020;89(2):41-54.

8. Golov R, Palamarchuk A, Anisimov K, Andrianov AJRER. Cluster policy in a digital economy. 2021;41:631-3.

9. Notley TJMIA. The environmental costs of the global digital economy in Asia and the urgent need for better policy. 2019;173(1):125-41.

10. Van Gorp N, Honnefelder SJC, Strategies. Challenges for competition policy in the digitalised economy. 2015;(99):149.

11. Ruiz Estrada MAJAaS. The policy modeling research consistency index (PMC-index). 2010.

12. Jiang XJJoCE, Studies B. Digital economy in the post-pandemic era. 2020;18(4):333-9.

13. Zhou G. The Impact of COVID-19 Pandemic on the Digital Economy and Policy Responses. COVID-19's Economic Impact and Countermeasures in China2022. p. 175-97.

14. Liu Y, Li J, Xu Y. Quantitative Evaluation of High-Tech Industry Policies Based on the PMC-Index Model: A Case Study of China’s Beijing-Tianjin-Hebei Region. Sustainability. 2022;14(15). doi: 10.3390/su14159338.

15. Dai S, Zhang W, Zong J, Wang Y, Wang G. How Effective Is the Green Development Policy of China’s Yangtze River Economic Belt? A Quantitative Evaluation Based on the PMC-Index Model. International Journal of Environmental Research and Public Health. 2021;18(14). doi: 10.3390/ijerph18147676.

16. Kuang B, Han J, Lu X, Zhang X, Fan X. Quantitative evaluation of China’s cultivated land protection policies based on the PMC-Index model. Land Use Policy. 2020;99. doi: 10.1016/j.landusepol.2020.105062.

17. Lu C, Wang B, Chen T, Yang J. A Document Analysis of Peak Carbon Emissions and Carbon Neutrality Policies Based on a PMC Index Model in China. Int J Environ Res Public Health. 2022;19(15). Epub 2022/08/13. doi: 10.3390/ijerph19159312. PubMed PMID: 35954666; PubMed Central PMCID: PMCPMC9368600.

18. Nkoua Nkuika GLF, Yiqun X. Quantitative Evaluation and Optimization Path of Advanced Manufacturing Development Policy Based on the PMC–AE Index Model. International Journal of Global Business and Competitiveness. 2022;17(S1):1-11. doi: 10.1007/s42943-022-00063-x.

19. Liu L, Chen J, Wang C, Wang Q. Quantitative evaluation of China's basin ecological compensation policies based on the PMC index model. Environ Sci Pollut Res Int. 2023;30(7):17532-45. Epub 2022/10/06. doi: 10.1007/s11356-022-23354-5. PubMed PMID: 36197610.

20. Zhao Y, Wu L. Research on Emergency Response Policy for Public Health Emergencies in China-Based on Content Analysis of Policy Text and PMC-Index Model. Int J Environ Res Public Health. 2022;19(19). Epub 2022/10/15. doi: 10.3390/ijerph191912909. PubMed PMID: 36232209; PubMed Central PMCID: PMCPMC9566489.

21. Liu F, Liu Z. Quantitative Evaluation of Waste Separation Management Policies in the Yangtze River Delta Based on the PMC Index Model. Int J Environ Res Public Health. 2022;19(7). Epub 2022/04/13. doi: 10.3390/ijerph19073815. PubMed PMID: 35409497; PubMed Central PMCID: PMCPMC8998125.

22. Li Z, Guo X. Quantitative evaluation of China's disaster relief policies: A PMC index model approach. International Journal of Disaster Risk Reduction. 2022;74. doi: 10.1016/j.ijdrr.2022.102911.

23. Li Y, He R, Liu J, Li C, Xiong J. Quantitative Evaluation of China’s Pork Industry Policy: A PMC Index Model Approach. Agriculture. 2021;11(2). doi: 10.3390/agriculture11020086.

24. Khan A, Yusof Z. Terrorist economic impact evaluation (TEIE) model: the case of Pakistan. Quality & Quantity. 2016;51(3):1381-94. doi: 10.1007/s11135-016-0336-z.

25. Estrada MARJJoPM. Policy modeling: Definition, classification and evaluation. 2011;33(4):523-36.

26. Liu J, Li N, Cheng L. Mining and quantitative evaluation of COVID-19 policy tools in China. PLoS One. 2023;18(4):e0284143. Epub 2023/04/08. doi: 10.1371/journal.pone.0284143. PubMed PMID: 37027438; PubMed Central PMCID: PMCPMC10081750.

27. Yang T, Xing C, Li X. Evaluation and analysis of new-energy vehicle industry policies in the context of technical innovation in China. Journal of Cleaner Production. 2021;281. doi: 10.1016/j.jclepro.2020.125126.

Reviewer #2: 

This manuscript studied quantitative evaluation of digital economy policy in China based on the PMC index model. Although the topic is interesting, the innovation and contribution is not high. In addition, there are some shortcomings in this manuscript. Firslty, variable setting of PMC model of digital economy policy is not clearly. Please illustrate in detail. Secondly, how to testify the efficency of the PMC model? Can you compare with other models? Finally, the corresponding countermeasures and suggestions should be more concrete and can be executed.

Response: Thank you for the feedback from the reviewing expert. In this study, all the primary variables of the PMC index model for digital economy policy are based on the research of scholars such as Ruiz, Liu, and Yang. A total of 10 primary variables have been extracted: Policy Type (X1), Policy Effectiveness (X2), Policy Level (X3), Policy Felids (X4), Policy Guarantee (X5), Policy Audience (X6), Policy Objectives (X7), Policy Evaluation (X8), Policy Perspective (X9), and Policy Publicity (X10). Due to the nature of policy, policy tools, and other primary variables, the underlying secondary variables cannot be directly obtained from policy texts. Therefore, the use of text mining tool Rostcm.6 is necessary for conducting word frequency analysis. Based on the characteristics of digital economic development, a total of 10 primary indicators and 45 secondary indicators have been constructed.

The PMC index model follows a binary principle, where a value of 1 indicates the presence of a factor, and 0 indicates its absence. The reason for choosing this model is based on the consideration that this approach avoids overlooking any variables, treating each variable's importance consistently. Additionally, the paper highlights the advantages and disadvantages of other policy evaluation models, leading to the final selection of the PMC index evaluation model.

The theory of public policy evaluation is a theoretical framework for assessing policies, aimed at determining their effectiveness, feasibility, sustainability, and fairness. Its scope of application encompasses various domains, including government policies, social policies, and economic policies. On one hand, this theory enhances the scientific and practical aspects of policy formulation by offering a scientific and systematic approach to evaluating policy effects. On the other hand, evaluation outcomes assist policymakers in comprehending the effects of policy implementation, thereby enabling policy adjustments and improvements based on the evaluation results. Existing methods for public policy evaluation each possess distinctive characteristics, providing crucial references for assessments related to digital economic policies. Due to the distinct scopes of application for each method, evaluation outcomes also exhibit slight variations. The grey relational degree model exhibits subjectivity and substantial error in determining optimal values for various indicators. The slow convergence of the BP neural network hampers the precision of policy evaluation. The fuzzy comprehensive evaluation method lacks objectivity in assessing indicators. It's noteworthy that current policy evaluation research excessively emphasizes ex-post evaluation, paying insufficient attention to the policies themselves. In contrast, the PMC (Policy Modeling Consistency) index model showcases a more prominent comprehensive evaluation effect. This method underscores the internal consistency of policies, offering advantages such as diversity assessment, pliability of indicators, and distinguishable gradations. Additionally, serving as a model for evaluating policy texts, it boasts traits like low cost and ease of operation, which mitigate subjectivity during the evaluation process. While assessing the merits and demerits of individual policies, it better reveals policy disparities across different periods or regions through policy comparisons.

Finally, based on the feedback from the peer reviewers, this study has incorporated and enhanced the recommendations and strategies. The specific improvements are as follows:

Research Limitations and Future Trends

It should be noted that this study primarily analyzed national and provincial policies in China and did not select city-level policies within various provinces. Due to variations in the level of digital industry development among different municipalities, a unified calculation standard has not yet been established. Therefore, future research should aim to expand the sample size and diversify the selection of subjects. Additionally, a comprehensive cross-analysis of policy content can be conducted from different dimensions by integrating policy tool theories. This approach can offer decision-makers objective and feasible solutions. Regarding policy quantification assessment, experimenting with various evaluation methods and selecting the optimal option after comprehensive comparisons could be attempted.

Building upon the research findings of this paper and considering the current development of China's digital economic industry, future studies can be directed towards the following aspects:

On one hand, it is essential to accelerate the integration of the digital economy with the real economy. As the digital economy and its systems become increasingly sophisticated, policies need to be appropriately adjusted, encompassing goals, positioning, and tools. Focusing on digital industrialization and industry digitization, policies should provide broader application scenarios for digital economic development, amplifying and multiplying the impact of digital technologies on economic growth.

On the other hand, emphasis should be placed on innovation-driven development. Presently, there are noticeable disparities in the development of the digital economy across the eastern, central, and western regions of China. Innovation-driven approaches can bridge these regional gaps, serving as a crucial lever for regional coordination. This strategy enables the utilization of developed digital economic regions' driving roles and innovation advantages, enhances regional collaborative innovation capabilities, and promotes high-quality development of the digital economic industry.

Lastly, in order to achieve the policy objectives of digital industrialization and industry digitization, internal demand needs to be constantly improved and external demand needs to be fully utilized. The development of digital economy is long-term and systematic. Therefore, it is needed to accelerate the establishment of data element market, promote the equality and benefits of public services, improve the governance system of the digital economy, eliminate the digital divide and lay a solid foundation for building a good modern market system of digital economy.

Reviewer #3: 

First of all, I congratulates to the authors for making a significant empirical contribution in economic literature. The author implemented significant contribution in order to have their work published in the PLOS One journal.

Response: Thank you for the recognition and trust of the reviewing experts.

Sincerely,

Tianzun Wang

---

## [Decision Letter · Decision Letter 1]

24 Nov 2023

PONE-D-23-07522R1Research on Quantitative Evaluation of Digital Economy Policy in China Based on the PMC Index ModelPLOS ONE

Dear Dr. WANG,

Thank you for submitting your manuscript to PLOS ONE. After careful consideration, we feel that it has merit but does not fully meet PLOS ONE’s publication criteria as it currently stands. Therefore, we invite you to submit a revised version of the manuscript that addresses the points raised during the review process.

Your manuscript has been evaluated by four reviewers: two prior reviewers (Reviewers 1 and 3) and two new reviewers (Reviewers 4 and 5); their comments are appended below. The reviewers have requested further revisions, particularly on clarity regarding the exact contribution this work adds to the literature on this topic, the selection criteria for the policy texts analyzed, and the rationale for the variables selected. This is in addition to further methodological details regarding the PMC Index Model used and  explanation of the formulas mentioned. Please ensure you address each of the reviewers' comments when revising your manuscript.

We look forward to receiving your revised manuscript.

Kind regards,

Hugh Cowley

Staff Editor

PLOS ONE

Reviewers' comments:

Reviewer's Responses to Questions

**Comments to the Author**

1. If the authors have adequately addressed your comments raised in a previous round of review and you feel that this manuscript is now acceptable for publication, you may indicate that here to bypass the “Comments to the Author” section, enter your conflict of interest statement in the “Confidential to Editor” section, and submit your "Accept" recommendation.

Reviewer #1: All comments have been addressed

Reviewer #3: All comments have been addressed

Reviewer #4: All comments have been addressed

Reviewer #5: All comments have been addressed

2. Is the manuscript technically sound, and do the data support the conclusions?

Reviewer #1: No

Reviewer #3: Yes

Reviewer #4: Yes

Reviewer #5: Yes

3. Has the statistical analysis been performed appropriately and rigorously? 

Reviewer #1: No

Reviewer #3: Yes

Reviewer #4: Yes

Reviewer #5: N/A

4. Have the authors made all data underlying the findings in their manuscript fully available?

Reviewer #1: No

Reviewer #3: Yes

Reviewer #4: Yes

Reviewer #5: Yes

5. Is the manuscript presented in an intelligible fashion and written in standard English?

Reviewer #1: No

Reviewer #3: Yes

Reviewer #4: Yes

Reviewer #5: Yes

6. Review Comments to the Author

Reviewer #1: 1. In order to establish the significance and value of the study, it is necessary to provide a comprehensive rationale for the research, which emphasizes its relevance and unique contributions to the current scholarly discourse. This will strengthen the study's originality and scholarly impact. Author don’t provide clearly? revised it

2. The authors are requested to provide a thorough justification for the choice of the current methodology, including an explanation of the benefits and advantages it offers over alternative approaches. This will lend greater credibility and rigor to the study's research design.

3. To highlight the novelty and contribution of the current study, it is necessary to outline the ways in which it diverges from the existing body of literature. Therefore, the concluding section of the literature review should include a clear differentiation of the study from prior research.

4. I would kindly request that the authors re-examine the policy implications derived from the study's findings, to ensure their accuracy and relevance to the study's objectives.

5. In order to ensure the relevance and timeliness of the study’s cited sources, it is recommended that the authors review and update the references section.

6. The study appears to employ an excessive amount of acronyms, which may impede comprehension for the wider readership of the journal. Thus, it is recommended that the authors avoid excessive usage of acronyms and instead use the full form of terms to ensure ease of understanding for readers.

7. The paper examines such selecting the text of digital economy policy issued by China government, the paper constructs a quantitative evaluation model of digital economy policy using the methods of content analysis and text mining etc. What is the unified theoretical framework for the impact of evaluation model of digital economy policy? able to combine all the variable above. What is the theoretical basis for the integration of multiple factors? In addition, why not put the above dependent and independent variables into a unified.

Reviewer #3: I congratulates to the authors for making a significant empirical

contribution in economic literature. The author implemented significant contribution in

order to have their work published in the PLOS One journal.

Reviewer #4: (No Response)

Reviewer #5: Research on Quantitative Evaluation of Digital Economy Policy in China Based on the PMC Index Model

The paper presents a quantitative evaluation of digital economy policies in China utilizing the PMC Index Model. While the study provides valuable insights into the strengths and weaknesses of China's digital economy policies, there are several areas that require improvement to enhance the overall quality and impact of the research. while the paper offers valuable insights into the evaluation of digital economy policies in China, addressing the mentioned areas of improvement is crucial. Enhancing the clarity of the methodology, providing a strong theoretical framework, detailing the variables' selection rationale, and offering practical policy implications will significantly enhance the paper's overall quality and impact. These revisions would strengthen the paper's contribution to the field of digital economy policy research and make it more accessible to a wider readership.

• The literature review provides relevant background information on the digital economy; however, it could be strengthened by incorporating more recent and diverse sources. Additionally, a critical analysis of existing literature, pinpointing the gaps the current study aims to fill, would add depth to the review section.

• The paper references the use of 37 policy texts but lacks information on the selection criteria for these texts. Providing clear details on the criteria used for policy selection would enhance the transparency and credibility of the research.

• The paper includes a comprehensive list of variables used in the evaluation model. However, it is essential to explain the rationale behind selecting these specific variables. Clarifying why these variables were chosen and how they align with the study's objectives would strengthen the research methodology.

• The paper employs the PMC Index Model for policy evaluation, which is a robust approach. However, the technical complexity of the model requires a more detailed and explanation. Providing a clear rationale for the model's construction.

• The formulas (1)-(4) used for calculations are mentioned but lack detailed explanation. A thorough breakdown of these formulas and their significance is necessary for readers to grasp the calculations and interpretations fully.

• Information regarding the sources of data used for the analysis is missing, making it challenging to assess the reliability of the results. Additionally, it is crucial to discuss limitations related to the data sources and methodology to present a balanced view of the study's scope and potential constraints.

• on policy recommendations, a more in-depth discussion of the implications of the findings for policymakers, businesses, and other stakeholders is needed. Exploring practical applications and potential strategies based on the research findings would enhance the paper's relevance and practicality.

7. PLOS authors have the option to publish the peer review history of their article (what does this mean?). If published, this will include your full peer review and any attached files.

Reviewer #1: **Yes: **KASHIF ABBASS

Reviewer #3: **Yes: **Dr shabbir ahmed department of economics Government islamia graduate college kasur pakistan

Reviewer #4: No

Reviewer #5: No

---

## [Author Response · Author response to Decision Letter 1]

8 Jan 2024

Dear Editor and Reviewers.

Thank you for offering us an opportunity to improve the quality of our submitted manuscript (PONE-D-23-07522R1). We appreciated very much the reviewers’ constructive and insightful comments. In this revision, we have addressed all of these comments. We hope the revised manuscript has now met the publication standard of your journal.

We highlighted all the revisions in yellow colour.

On the next pages, our point-to-point responses to the queries raised by the reviewers are listed.

Reviewer #1:

1. In order to establish the significance and value of the study, it is necessary to provide a comprehensive rationale for the research, which emphasizes its relevance and unique contributions to the current scholarly discourse. This will strengthen the study's originality and scholarly impact. Author don’t provide clearly? revised it

Response: Regarding the comprehensive rationale for the research, we have supplemented and modified it. The specific modifications are as follows:

Compared with other methods, the PMC index model has important advantages. The evaluation dimensions are rich, and by adding evaluation dimensions instead of calculating indicator weights, it effectively avoids indicator weight errors and subjective evaluation biases, making the evaluation results more objective and accurate. Focusing on pre-policy evaluation, analyzing key content and keywords in policy texts, fills the gap in research and analysis of policy content. The PMC index evaluation method can be applied to both single industry policies and regional policy systems.

The digital economy policy is a series of policy tool combinations that promote the development of the digital economy, involving multiple dimensions such as policy nature, policy objectives, policy content, policy effects, etc. How to objectively evaluate is a very important issue. At present, there is relatively little literature on the evaluation of digital economy policies, and policy evaluation is the most important link in the process of public policy formulation and management [25, 26]. It uses relevant research methods to systematically measure and judge the effectiveness of policy intervention and implementation [27]. Through the evaluation of digital economy policies, not only can scientific judgments be made on the value of the policies themselves [28], but also the actual effects of policy formulation and implementation can be tested [29]. Therefore, in view of this, this article combines digital economy policy sample analysis, text mining, and PMC index model to construct a quantitative evaluation index system for digital economy policies. It conducts sample analysis and text mining on China's digital economy policies, and quantitatively evaluates and analyzes typical digital economy policy texts at the central and local levels, in order to provide decision-making basis for the improvement of relevant policies and the formulation of new policies.

25. Abbass K, Song H, Khan F, Begum H, Asif M. Fresh insight through the VAR approach to investigate the effects of fiscal policy on environmental pollution in Pakistan. Environ Sci Pollut Res Int. 2022;29(16):23001-14. Epub 2021/11/20. doi: 10.1007/s11356-021-17438-x. PubMed PMID: 34797543.

26. Abbass K, Qasim MZ, Song H, Murshed M, Mahmood H, Younis I. A review of the global climate change impacts, adaptation, and sustainable mitigation measures. Environmental Science and Pollution Research. 2022;29(28):42539-59. doi: 10.1007/s11356-022-19718-6.

27. Abbass K, Niazi AAK, Basit A, Qazi TF, Song H, Begum H. Uncovering Effects of Hot Potatoes in Banking System: Arresting Die-Hard Issues. SAGE Open. 2021;11(4). doi: 10.1177/21582440211061554.

28. Fu H, Abbass K, Qazi TF, Niazi AAK, Achim MV. Analyzing the barriers to putting corporate financial expropriations to a halt: A structural modeling of the phenomenon. Frontiers in Environmental Science. 2022;10. doi: 10.3389/fenvs.2022.967418.

29. Amin N, Shabbir MS, Song H, Abbass K. Renewable energy consumption and its impact on environmental quality: A pathway for achieving sustainable development goals in ASEAN countries. Energy & Environment. 2022. doi: 10.1177/0958305x221134113.

Any single indicator can be misleading, but if multiple composite indicators point to the same result, it can provide a more accurate judgment of the evaluated thing. The PMC index first determines the meaning and level of variables at all levels, and then evaluates and analyzes the policy's advantages and disadvantages through the aggregated consistency level. This method attempts to find highly saturated secondary variables that can characterize policy characteristics and assign consistent weights to these variables to avoid subjective limitations. In addition, all variables are binary balanced, greatly simplifying the complexity of PMC index calculation.

This article uses a composite analysis method that combines policy sample analysis, text mining, and PMC index model. On the basis of sorting out China's digital economy policies, text mining is used to search for important and highly correlated text data, which forms a component of secondary indicators. Unstructured text data is transformed into structured and readable data, and the PMC index model is used to conduct quantitative evaluation research on digital economy policies in central and local areas of China.

Compared with the traditional economy, the digital economy focuses on the application of products and the extension of services, is demand oriented, focuses on discovering potential and intangible user needs, provides personalized services, and creates user value. Due to the existence of digital technology, more consumers are involved in the production and consumption of products, so the dominant position of the digital economy is relatively unclear.

The digital economy is essentially a technology economy paradigm, which is an optimal practice model for economy and society based on technological innovation. It responds to structural crises caused by technological changes through institutional changes, thus forming a relatively stable and sustainable behavior. The digital economy is driven by digital knowledge and information as key production factors, modern information networks as the main carrier, and the efficient utilization of information and communication technology as an important driving force for efficiency improvement and economic optimization. Therefore, introducing data elements and changing social production methods can provide an important experimental environment for expanding current economic and management theories; The development of the digital economy requires new theories from economics and management to explain and promote the construction of new theories.

Since China first included the digital economy in the Two Sessions Report in 2017, the national level has attached increasing importance to promoting the development of the digital economy. To address various practical issues in the development of the digital economy, various Chinese ministries have issued a series of policies and regulations. The introduction of numerous policies has also made the policy and regulatory system in this field increasingly complex, leading to ineffective implementation and poor coordination in the policy integration environment. Policy analysis is the main basis for the abolition, reform, and establishment of digital economy policies. The primary task of policy analysis is to identify and conceptualize the problems that need to be solved, and policy problems stem from unresolved practical problems within the current policy system. The evaluation and analysis based on digital economy policy texts have important practical value and significance. Therefore, when quantitatively evaluating China's digital economy policies, this article will focus on distinguishing the advantages and disadvantages of current policies based on existing practical problems, providing decision-making basis for the improvement of relevant policies and the formulation of new policies.

2. The authors are requested to provide a thorough justification for the choice of the current methodology, including an explanation of the benefits and advantages it offers over alternative approaches. This will lend greater credibility and rigor to the study's research design.

Response: Thanks for your suggestion. Regarding the alternative approaches, we have supplemented and modified it. The specific modifications are as follows:

Compared with other methods, the PMC index model has important advantages. The evaluation dimensions are rich, and by adding evaluation dimensions instead of calculating indicator weights, it effectively avoids indicator weight errors and subjective evaluation biases, making the evaluation results more objective and accurate. Focusing on pre-policy evaluation, analyzing key content and keywords in policy texts, fills the gap in research and analysis of policy content. The PMC index evaluation method can be applied to both single industry policies and regional policy systems.

3. To highlight the novelty and contribution of the current study, it is necessary to outline the ways in which it diverges from the existing body of literature. Therefore, the concluding section of the literature review should include a clear differentiation of the study from prior research.

Response: Thanks for your suggestion. Regarding the differentiation of the study from prior research, we have supplemented and modified it. The specific modifications are as follows:

The digital economy policy is a series of policy tool combinations that promote the development of the digital economy, involving multiple dimensions such as policy nature, policy objectives, policy content, policy effects, etc. How to objectively evaluate is a very important issue. At present, there is relatively little literature on the evaluation of digital economy policies, and policy evaluation is the most important link in the process of public policy formulation and management [25, 26]. It uses relevant research methods to systematically measure and judge the effectiveness of policy intervention and implementation [27]. Through the evaluation of digital economy policies, not only can scientific judgments be made on the value of the policies themselves [28], but also the actual effects of policy formulation and implementation can be tested [29]. Therefore, in view of this, this article combines digital economy policy sample analysis, text mining, and PMC index model to construct a quantitative evaluation index system for digital economy policies. It conducts sample analysis and text mining on China's digital economy policies, and quantitatively evaluates and analyzes typical digital economy policy texts at the central and local levels, in order to provide decision-making basis for the improvement of relevant policies and the formulation of new policies.

25. Abbass K, Song H, Khan F, Begum H, Asif M. Fresh insight through the VAR approach to investigate the effects of fiscal policy on environmental pollution in Pakistan. Environ Sci Pollut Res Int. 2022;29(16):23001-14. Epub 2021/11/20. doi: 10.1007/s11356-021-17438-x. PubMed PMID: 34797543.

26. Abbass K, Qasim MZ, Song H, Murshed M, Mahmood H, Younis I. A review of the global climate change impacts, adaptation, and sustainable mitigation measures. Environmental Science and Pollution Research. 2022;29(28):42539-59. doi: 10.1007/s11356-022-19718-6.

27. Abbass K, Niazi AAK, Basit A, Qazi TF, Song H, Begum H. Uncovering Effects of Hot Potatoes in Banking System: Arresting Die-Hard Issues. SAGE Open. 2021;11(4). doi: 10.1177/21582440211061554.

28. Fu H, Abbass K, Qazi TF, Niazi AAK, Achim MV. Analyzing the barriers to putting corporate financial expropriations to a halt: A structural modeling of the phenomenon. Frontiers in Environmental Science. 2022;10. doi: 10.3389/fenvs.2022.967418.

29. Amin N, Shabbir MS, Song H, Abbass K. Renewable energy consumption and its impact on environmental quality: A pathway for achieving sustainable development goals in ASEAN countries. Energy & Environment. 2022. doi: 10.1177/0958305x221134113.

4. I would kindly request that the authors re-examine the policy implications derived from the study's findings, to ensure their accuracy and relevance to the study's objectives.

Response: Thanks for your suggestion. Regarding the policy implications on the role of stakeholders in the digital economy, we have supplemented and modified it. The specific modifications are as follows:

Finally, pay attention to the role of stakeholders in the digital economy. In the era of digital economy, stakeholders have already extended to employees, users, and partners. Stakeholders should not only include shareholders, but also employees in the future. We should not only focus on shareholders, but also on employees. Only with excellent employees can we create greater value for the enterprise and achieve better development. The digital economy is built on the basis of data, and enterprises earn more profits by collecting user data. Enterprises should be regulated while using data to earn more profits. Regulators need to play a more important role in protecting user privacy and data usage. Protecting privacy requires stricter regulations to ensure it. The government is facing a new market composed of an increasing number of platforms, which poses an unprecedented challenge to its market governance. On the basis of a scientific understanding of the new economic form of the digital economy and a rational analysis of the two growth paths of the digital economy, the government should redesign the content of government management functions in the digital economy and further play a positive role.

5. In order to ensure the relevance and timeliness of the study’s cited sources, it is recommended that the authors review and update the references section.

Response: Thanks for your suggestion. We have updated and improved the content of the references section. The modifications are as follows:

14. Abbass K, Begum H, Alam ASAF, Awang AH, Abdelsalam MK, Egdair IMM, et al. Fresh Insight through a Keynesian Theory Approach to Investigate the Economic Impact of the COVID-19 Pandemic in Pakistan. Sustainability. 2022;14(3). doi: 10.3390/su14031054.

17. Liu F, Tang J, Rustam A, Liu Z. Evaluation of the central and local power batteries recycling policies in China: A PMC-Index model approach. Journal of Cleaner Production. 2023;427. doi: 10.1016/j.jclepro.2023.139073.

25. Abbass K, Song H, Khan F, Begum H, Asif M. Fresh insight through the VAR approach to investigate the effects of fiscal policy on environmental pollution in Pakistan. Environ Sci Pollut Res Int. 2022;29(16):23001-14. Epub 2021/11/20. doi: 10.1007/s11356-021-17438-x. PubMed PMID: 34797543.

26. Abbass K, Qasim MZ, Song H, Murshed M, Mahmood H, Younis I. A review of the global climate change impacts, adaptation, and sustainable mitigation measures. Environmental Science and Pollution Research. 2022;29(28):42539-59. doi: 10.1007/s11356-022-19718-6.

27. Abbass K, Niazi AAK, Basit A, Qazi TF, Song H, Begum H. Uncovering Effects of Hot Potatoes in Banking System: Arresting Die-Hard Issues. SAGE Open. 2021;11(4). doi: 10.1177/21582440211061554.

28. Fu H, Abbass K, Qazi TF, Niazi AAK, Achim MV. Analyzing the barriers to putting corporate financial expropriations to a halt: A structural modeling of the phenomenon. Frontiers in Environmental Science. 2022;10. doi: 10.3389/fenvs.2022.967418.

29. Amin N, Shabbir MS, Song H, Abbass K. Renewable energy consumption and its impact on environmental quality: A pathway for achieving sustainable development goals in ASEAN countries. Energy & Environment. 2022. doi: 10.1177/0958305x221134113.

30. Xiong Y, Zhang C, Qi H. How effective is the fire safety education policy in China? A quantitative evaluation based on the PMC-index model. Safety Science. 2023;161. doi: 10.1016/j.ssci.2023.106070.

32. Kuang B, Han J, Lu X, Zhang X, Fan X. Quantitative evaluation of China’s cultivated land protection policies based on the PMC-Index model. Land Use Policy. 2020;99. doi: 10.1016/j.landusepol.2020.105062.

34. Hu M, Guo C, Wang Y, Ma D. Quantitative evaluation of China's private universities provincial public funding policies based on the PMC-Index model. PLoS One. 2023;18(12):e0295601. Epub 2023/12/12. doi: 10.1371/journal.pone.0295601. PubMed PMID: 38085719; PubMed Central PMCID: PMCPMC10715659.

36. Meng J, Xu W. Quantitative Evaluation of Carbon Reduction Policy Based on the Background of Global Climate Change. Sustainability. 2023;15(19). doi: 10.3390/su151914581.

37. Xu J, Zhang Z, Xu Y, Liu L, Pei T. Quantitative evaluation of waste sorting management policies in China’s major cities based on the PMC index model. Frontiers in Environmental Science. 2023;11. doi: 10.3389/fenvs.2023.1065900.

6. The study appears to employ an excessive amount of acronyms, which may impede comprehension for the wider readership of the journal. Thus, it is recommended that the authors avoid excessive usage of acronyms and instead use the full form of terms to ensure ease of understanding for readers.

Response: Thanks for your suggestion. We have provided definitions for terms such as Corona Virus Disease 2019 COVID-19 () and Back Propagation (BP) neural network legal database in their initial instances of use within the article. It should be noted that this study employed the Policy Modeling Consistency (PMC) index model for the quantitative assessment of digital economic policies. As a result, there are multiple instances of the term "PMC" throughout the text, but definitions have been provided at its initial use.

7. The paper examines such selecting the text of digital economy policy issued by China government, the paper constructs a quantitative evaluation model of digital economy policy using the methods of content analysis and text mining etc. What is the unified theoretical framework for the impact of evaluation model of digital economy policy? able to combine all the variable above. What is the theoretical basis for the integration of multiple factors? In addition, why not put the above dependent and independent variables into a unified.

Response: Thanks for your suggestion. Regarding the policy evaluation of digital economy , we have supplemented and modified it. The specific modifications are as follows:

Any single indicator can be misleading, but if multiple composite indicators point to the same result, it can provide a more accurate judgment of the evaluated thing. The PMC index first determines the meaning and level of variables at all levels, and then evaluates and analyzes the policy's advantages and disadvantages through the aggregated consistency level. This method attempts to find highly saturated secondary variables that can characterize policy characteristics and assign consistent weights to these variables to avoid subjective limitations. In addition, all variables are binary balanced, greatly simplifying the complexity of PMC index calculation.

This article uses a composite analysis method that combines policy sample analysis, text mining, and PMC index model. On the basis of sorting out China's digital economy policies, text mining is used to search for important and highly correlated text data, which forms a component of secondary indicators. Unstructured text data is transformed into structured and readable data, and the PMC index model is used to conduct quantitative evaluation research on digital economy policies in central and local areas of China.

Compared with the traditional economy, the digital economy focuses on the application of products and the extension of services, is demand oriented, focuses on discovering potential and intangible user needs, provides personalized services, and creates user value. Due to the existence of digital technology, more consumers are involved in the production and consumption of products, so the dominant position of the digital economy is relatively unclear.

The digital economy is essentially a technology economy paradigm, which is an optimal practice model for economy and society based on technological innovation. It responds to structural crises caused by technological changes through institutional changes, thus forming a relatively stable and sustainable behavior. The digital economy is driven by digital knowledge and information as key production factors, modern information networks as the main carrier, and the efficient utilization of information and communication technology as an important driving force for efficiency improvement and economic optimization. Therefore, introducing data elements and changing social production methods can provide an important experimental environment for expanding current economic and management theories; The development of the digital economy requires new theories from economics and management to explain and promote the construction of new theories.

Since China first included the digital economy in the Two Sessions Report in 2017, the national level has attached increasing importance to promoting the development of the digital economy. To address various practical issues in the development of the digital economy, various Chinese ministries have issued a series of policies and regulations. The introduction of numerous policies has also made the policy and regulatory system in this field increasingly complex, leading to ineffective implementation and poor coordination in the policy integration environment. Policy analysis is the main basis for the abolition, reform, and establishment of digital economy policies. The primary task of policy analysis is to identify and conceptualize the problems that need to be solved, and policy problems stem from unresolved practical problems within the current policy system. The evaluation and analysis based on digital economy policy texts have important practical value and significance. Therefore, when quantitatively evaluating China's digital economy policies, this article will focus on distinguishing the advantages and disadvantages of current policies based on existing practical problems, providing decision-making basis for the improvement of relevant policies and the formulation of new policies.

Reviewer #3:

I congratulates to the authors for making a significant empirical contribution in economic literature. The author implemented significant contribution in order to have their work published in the PLOS One journal.

Response: Thank you for the recognition and trust of the reviewing experts.

Reviewer #4: (No Response)

Reviewer #5:

1. The literature review provides relevant background information on the digital economy; however, it could be strengthened by incorporating more recent and diverse sources. Additionally, a critical analysis of existing literature, pinpointing the gaps the current study aims to fill, would add depth to the review section.

Response: Thanks for your suggestion. Regarding literature review, we have supplemented and modified it. The specific modifications are as follows:

The digital economy policy is a series of policy tool combinations that promote the development of the digital economy, involving multiple dimensions such as policy nature, policy objectives, policy content, policy effects, etc. How to objectively evaluate is a very important issue. At present, there is relatively little literature on the evaluation of digital economy policies, and policy evaluation is the most important link in the process of public policy formulation and management [25, 26]. It uses relevant research methods to systematically measure and judge the effectiveness of policy intervention and implementation [27]. Through the evaluation of digital economy policies, not only can scientific judgments be made on the value of the policies themselves [28], but also the actual effects of policy formulation and implementation can be tested [29]. Therefore, in view of this, this article combines digital economy policy sample analysis, text mining, and PMC index model to construct a quantitative evaluation index system for digital economy policies. It conducts sample analysis and text mining on China's digital economy policies, and quantitatively evaluates and analyzes typical digital economy policy texts at the central and local levels, in order to provide decision-making basis for the improvement of relevant policies and the formulation of new policies.

25. Abbass K, Song H, Khan F, Begum H, Asif M. Fresh insight through the VAR approach to investigate the effects of fiscal policy on environmental pollution in Pakistan. Environ Sci Pollut Res Int. 2022;29(16):23001-14. Epub 2021/11/20. doi: 10.1007/s11356-021-17438-x. PubMed PMID: 34797543.

26. Abbass K, Qasim MZ, Song H, Murshed M, Mahmood H, Younis I. A review of the global climate change impacts, adaptation, and sustainable mitigation measures. Environmental Science and Pollution Research. 2022;29(28):42539-59. doi: 10.1007/s11356-022-19718-6.

27. Abbass K, Niazi AAK, Basit A, Qazi TF, Song H, Begum H. Uncovering Effects of Hot Potatoes in Banking System: Arresting Die-Hard Issues. SAGE Open. 2021;11(4). doi: 10.1177/21582440211061554.

28. Fu H, Abbass K, Qazi TF, Niazi AAK, Achim MV. Analyzing the barriers to putting corporate financial expropriations to a halt: A structural modeling of the phenomenon. Frontiers in Environmental Science. 2022;10. doi: 10.3389/fenvs.2022.967418.

29. Amin N, Shabbir MS, Song H, Abbass K. Renewable energy consumption and its impact on environmental quality: A pathway for achieving sustainable development goals in ASEAN countries. Energy & Environment. 2022. doi: 10.1177/0958305x221134113.

2. The paper references the use of 37 policy texts but lacks information on the selection criteria for these texts. Providing clear details on the criteria used for policy selection would enhance the transparency and credibility of the research.

Response: Thank you for the reviewer's feedback. We have revised the selection sources for the 37 policy texts. The specific modifications are as follows:

The selection criteria for digital economic policies are as follows: To ensure the comprehensiveness and authority of the policy sample content, the focus is primarily on national and provincial-level digital economic policy texts. National-level policies, being overarching documents, provide stronger guidance and standardization, serving as crucial foundations for provincial policy formulation, guiding and constraining provincial policies. The selection of provincial policies considers their coordination with the central government and effectiveness. City-level and county-level policies are often extensions and supplements to provincial policies, and therefore, they are not included in the sample selection.

The initial policy search is conducted using the "PKU Law" (Peking University Legal Information Retrieval System) legal professional database with the title "Digital Economy," specifying the policy category as currently effective, and the retrieval date as January 1, 2023. Informal decision documents such as "letters" and "administrative licensing approvals" are excluded, focusing on formal decision documents like laws, regulations, resolutions, orders, opinions, and notifications. To ensure accuracy, verification and supplementation are performed on the official websites of the central government and various provincial governments. Finally, a manual full-text review is employed to eliminate policies with little relevance to the research theme. In total, 37 policy texts were retrieved (detailed information in Supporting Information S1 and S2), including 5 national-level policies and 32 provincial-level policies. The main distribution results are presented in Table 1.

3. The paper includes a comprehensive list of variables used in the evaluation model. However, it is essential to explain the rationale behind selecting these specific variables. Clarifying why these variables were chosen and how they align with the study's objectives would strengthen the research methodology.

Response: Thanks for your advice. Building upon previous research, this study has constructed 10 primary variables: Policy Type (X1), Policy Effectiveness (X2), Policy Level (X3), Policy Field (X4), Policy Guarantee (X5), Policy Audience (X6), Policy Objectives (X7), Policy Evaluation (X8), Policy Perspective (X9) and Policy Publicity (X10). Simultaneously, by combining the results of text mining with relevant scholars' studies, 45 secondary variables were formulated. The construction and interpretation of these variables are detailed and can be referred to in Table 3, "Variable Setting of PMC Model of Digital Economy Policy," along with the notes below Table 3.

Note: The formulation of Policy Type (X1) is based on reference [27]. The formulation of Policy Effectiveness (X2) is based on reference [28]. The formulation of Policy Level (X3) is based on reference [29]. The formulation of Policy Field (X4) is based on reference [30]. The formulation of Policy Guarantee (X5) is based on references [28], [29], and text mining results. The formulation of Policy Audience (X6) is based on references [28], [29], and text mining results. The formulation of Policy Objectives (X7) is based on references [28], [29], and text mining results. The formulation of Policy Evaluation (X8) is based on reference [30]. The formulation of Policy Perspective (X9) is based on reference [26].

4. The paper employs the PMC Index Model for policy evaluation, which is a robust approach. However, the technical complexity of the model requires a more detailed and explanation. Providing a clear rationale for the model's construction.

Response: Thanks for your suggestion. Regarding the PMC Index Model, we have supplemented and modified it. The specific modifications are as follows:

Any single indicator can be misleading, but if multiple composite indicators point to the same result, it can provide a more accurate judgment of the evaluated thing. The PMC index first determines the meaning and level of variables at all levels, and then evaluates and analyzes the policy's advantages and disadvantages through the aggregated consistency level. This method attempts to find highly saturated secondary variables that can characterize policy characteristics and assign consistent weights to these variables to avoid subjective limitations. In addition, all variables are binary balanced, greatly simplifying the complexity of PMC index calculation.

This article uses a composite analysis method that combines policy sample analysis, text mining, and PMC index model. On the basis of sorting out China's digital economy policies, text mining is used to search for important and highly correlated text data, which forms a component of secondary indicators. Unstructured text data is transformed into structured and readable data, and the PMC index model is used to conduct quantitative evaluation research on digital economy policies in central and local areas of China.

5. The formulas (1)-(4) used for calculations are mentioned but lack detailed explanation. A thorough breakdown of these formulas and their significance is necessary for readers to grasp the calculations and interpretations fully.

Response: Thank you for the suggestions from the reviewer. The detailed explanation for formulas (1) to (4) has been modified in the "Calculation of PMC index" section. The detailed modifications are as follows:

The calculation of the PMC index model revolves around four aspects [37]. First, inputting the primary and secondary variables from the previous text into a multi-input-output table according to Formula (1). Second, sequentially assigning values to the 45 secondary variables in the digital economic policy multi-input-output table using Formula (2). The values of the secondary variables follow a [0,1] distribution, with a value of 1 assigned if they meet the evaluation criteria and 0 if they do not. Third, calculating the values of the primary variables based on Formula (3). After assigning values to the secondary variables that follow a [0,1] distribution in the digital economic policy PMC index model, the sum of the secondary variable scores is obtained. Then, dividing the sum by the number of secondary variables contained in the respective primary variable yields the arithmetic mean, representing the value of that primary variable. Finally, using Formula (4) to sum the values of each primary variable in the digital economic policy PMC index model, the overall PMC index for each digital economic policy is obtained. The detailed calculation formulas are as follows:

37. Xu J, Zhang Z, Xu Y, Liu L, Pei T. Quantitative evaluation of waste sorting management policies in China’s major cities based on the PMC index model. Frontiers in Environmental Science. 2023;11. doi: 10.3389/fenvs.2023.1065900.

6. Information regarding the sources of data used for the analysis is missing, making it challenging to assess the reliability of the results. Additionally, it is crucial to discuss limitations related to the data sources and methodology to present a balanced view of the study's scope and potential constraints.

Response: Thanks for your suggestion. Regarding the information on data sources, we have supplemented and modified it. The specific modifications are as follows:

The selection criteria for digital economic policies are as follows: To ensure the comprehensiveness and authority of the policy sample content, the focus is primarily on national and provincial-level digital economic policy texts. National-level policies, being overarching documents, provide stronger guidance and standardization, serving as crucial foundations for provincial policy formulation, guiding and constraining provincial policies. The selection of provincial policies considers their coordination with the central government and effectiveness. City-level and county-level policies are often extensions and supplements to provincial policies, and therefore, they are not included in the sample selection.

The initial policy search is conducted using the "PKU Law" (Peking University Legal Information Retrieval System) legal professional database with the title "Digital Economy," specifying the policy category as currently effective, and the retrieval date as January 1, 2023. Informal decision documents such as "letters" and "administrative licensing approvals" are excluded, focusing on formal decision documents like laws, regulations, resolutions, orders, opinions, and notifications. To ensure accuracy, verification and supplementation are performed on the official websites of the central government and various provincial governments. Finally, a manual full-text review is employed to eliminate policies with little relevance to the research theme. In total, 37 policy texts were retrieved (detailed information in Supporting Information S1 and S2), including 5 national-level policies and 32 provincial-level policies. The main distribution results are presented in Table 1.

Regarding the discussion on research methods and limitations of data sources, we have made modifications in the section titled "Research Limitations and Future Trends." The detailed information is as follows:

It is important to note that this study primarily quantitatively evaluated China's national and provincial-level digital economic policies, excluding policies at the municipal and county levels. Due to variations in the level of digital industry development among different municipalities, a unified calculation standard has not yet been established. Therefore, future research should aim to expand the sample size and diversify the selection of subjects. Additionally, a comprehensive cross-analysis of policy content can be conducted from different dimensions by integrating policy tool theories. This approach can offer decision-makers objective and feasible solutions. Additionally, in terms of policy evaluation methods, since this study exclusively employed the PMC index model, the results are relatively singular. In the future, a more comprehensive approach should be considered, incorporating various evaluation methods rather than relying solely on a single approach.

7. on policy recommendations, a more in-depth discussion of the implications of the findings for policymakers, businesses, and other stakeholders is needed. Exploring practical applications and potential strategies based on the research findings would enhance the paper's relevance and practicality.

Response: Thanks for your suggestion. Regarding the information on the role of stakeholders in the digital economy, we have supplemented and modified it. The specific modifications are as follows:

Finally, pay attention to the role of stakeholders in the digital economy. In the era of digital economy, stakeholders have already extended to employees, users, and partners. Stakeholders should not only include shareholders, but also employees in the future. We should not only focus on shareholders, but also on employees. Only with excellent employees can we create greater value for the enterprise and achieve better development. The digital economy is built on the basis of data, and enterprises earn more profits by collecting user data. Enterprises should be regulated while using data to earn more profits. Regulators need to play a more important role in protecting user privacy and data usage. Protecting privacy requires stricter regulations to ensure it. The government is facing a new market composed of an increasing number of platforms, which poses an unprecedented challenge to its market governance. On the basis of a scientific understanding of the new economic form of the digital economy and a rational analysis of the two growth paths of the digital economy, the government should redesign the content of government management functions in the digital economy and further play a positive role.

Sincerely,

Tianzun Wang

---

## [Decision Letter · Decision Letter 2]

23 Jan 2024

Research on Quantitative Evaluation of Digital Economy Policy in China Based on the PMC Index Model

PONE-D-23-07522R2

Dear Dr. WANG,

We’re pleased to inform you that your manuscript has been judged scientifically suitable for publication and will be formally accepted for publication once it meets all outstanding technical requirements.

Kind regards,

Dr. Jiachao Peng

Academic Editor

PLOS ONE

Additional Editor Comments (optional):

Reviewers' comments:

Reviewer's Responses to Questions

**Comments to the Author**

1. If the authors have adequately addressed your comments raised in a previous round of review and you feel that this manuscript is now acceptable for publication, you may indicate that here to bypass the “Comments to the Author” section, enter your conflict of interest statement in the “Confidential to Editor” section, and submit your "Accept" recommendation.

Reviewer #1: All comments have been addressed

Reviewer #4: All comments have been addressed

Reviewer #5: All comments have been addressed

2. Is the manuscript technically sound, and do the data support the conclusions?

Reviewer #1: Yes

Reviewer #4: Yes

Reviewer #5: Yes

3. Has the statistical analysis been performed appropriately and rigorously? 

Reviewer #1: Yes

Reviewer #4: Yes

Reviewer #5: Yes

4. Have the authors made all data underlying the findings in their manuscript fully available?

Reviewer #1: Yes

Reviewer #4: Yes

Reviewer #5: Yes

5. Is the manuscript presented in an intelligible fashion and written in standard English?

Reviewer #1: Yes

Reviewer #4: Yes

Reviewer #5: Yes

6. Review Comments to the Author

Reviewer #1: I am glad to review this paper " Research on Quantitative Evaluation of Digital Economy Policy in China Based on the PMC Index Model" . I accept this paper for publication

Reviewer #4: The authors have substantially improved the manuscript. It can now be accepted for publication.

Congratulations to the authors.

Reviewer #5: The author has successfully revised the manuscript. While the English language in the paper is commendable, it could be improved further to meet standard requirements. The analysis is appropriate.

7. PLOS authors have the option to publish the peer review history of their article (what does this mean?). If published, this will include your full peer review and any attached files.

Reviewer #1: No

Reviewer #4: No

Reviewer #5: No

---

## [Editor Report · Acceptance letter]

6 Feb 2024

PONE-D-23-07522R2 

PLOS ONE

Dear Dr. Wang, 

I'm pleased to inform you that your manuscript has been deemed suitable for publication in PLOS ONE. Congratulations! Your manuscript is now being handed over to our production team.

Kind regards, 

on behalf of

Dr. Jiachao Peng 

Academic Editor

PLOS ONE